

# 1     Phosphorus addition mitigates $N_2O$ and $CH_4$ emissions in N-
# 2     saturated subtropical forest, SW China

Longfei Yu[1], Yihao Wang[2, 3], Xiaoshan Zhang[3], Peter Dörsch[1], Jan Mulder[1*]
[1]Department of Environmental Sciences, Norwegian University of Life Sciences, Postbox 5003,
N-1432 Aas, Norway.
[2]Chongqing Academy of Forestry, 400036, Chongqing, China.
[3]Research Center for Eco-Environmental Sciences, Chinese Academy of Sciences, 100085,
Beijing, China
[*]Correspondence: Jan Mulder, tel. +47 67231852, E-mail jan.mulder@nmbu.no
Article type: Research Article





**Abstract**
Chronically elevated nitrogen (N) deposition has led to severe nutrient imbalance in forest soils.
Particularly in tropical and subtropical forest ecosystems, increasing N loading has aggravated
phosphorus (P) limitation of biomass production, and has resulted in elevated emissions of
nitrous oxide ($N_2O$) and reduced uptake of methane ($CH_4$), both of which are important
greenhouse gases. Yet, the interactions of N and P and their effects on GHG emissions remain
understudied. Here, we report $N_2O$ and $CH_4$ emissions together with soil chemistry data for the a
period of 18 months following P addition (79 kg P ha$^{-1}$ yr$^{-1}$, applied as $NaH_2PO_4$ powder) to a N-
saturated, Masson pine-dominated forest at TieShanPing (TSP), Chongqing, SW China. We
observed a significant decline both in $NO_3^-$ concentrations in soil water (at 5- and 20-cm depths)
and in $N_2O$ emissions, the latter by 3 kg N ha$^{-1}$ yr$^{-1}$. We hypothesize that enhanced N uptake by
plants and soil microbes in response to P addition, results in less available $NO_3^-$ for
denitrification. By contrast to most other forest ecosystems, TSP is a net source of $CH_4$. As for
$N_2O$, P addition significantly decreased $CH_4$ emissions, turning the soil into a net sink. Based on
our data and previous studies in South America and China, we believe that P addition relieves N-
inhibition of $CH_4$ oxidation. Within the 1.5 years after P addition, no significant increase of
forest growth was observed at TSP, but we cannot exclude that understory vegetation increased.
Our study suggests that P fertilization of acid forest soils could mitigate GHG emissions in
addition to alleviate nutrient imbalances and reduce losses of nitrogen through $NO_3^-$ leaching and
$N_2O$ emission.
**Key Word:** $N_2O$ and $CH_4$ emission, N saturation, Phosphate fertilization, soil $CH_4$ uptake, acid
forest soil.



## 1 Introduction

Anthropogenic activities have transformed the terrestrial biosphere into a net source of $CH_4$, $N_2O$ and $CO_2$, leading to increased radiative forcing (Montzka et al., 2011; Tian et al., 2016). During the last decade, atmospheric concentrations of $CO_2$, $CH_4$, $N_2O$ have increased at rates of 1.9 ppm $yr^{-1}$, 4.8 and 0.8 ppb $yr^{-1}$, respectively (Hartmann et al., 2013). In China, the exponential increase of reactive nitrogen (N) input into the biosphere since the 1970s has likely led to more carbon (C) being sequestered in the biosphere (Cui et al., 2013; Shi et al., 2015). However, enhanced emissions of $N_2O$ and $CH_4$ due to chronic N pollution potentially offset the cooling effect by C sequestration (Liu and Greaver, 2009; Tian et al., 2011).

Microbial nitrification and denitrification in soils account for about 60% of $N_2O$ emissions globally (Ciais et al., 2013; Hu et al., 2015). Although, microbial activity is often restricted in low pH soils of unproductive forests, surprisingly large $N_2O$ emissions have been reported from acid, upland forest soils in South China (Zhu et al., 2013b). Reported average $N_2O$ fluxes in humid, subtropical forests range from 2.0 to 5.4 kg $ha^{-1}$ $yr^{-1}$ (Fang et al., 2009; Tang et al., 2006; Zhu et al., 2013b), which by far exceeds global averages for temperate or tropical forest ecosystems (Werner et al., 2007; Zhuang et al., 2012). This has been attributed to frequently shifting aeration conditions during monsoonal summers, promoting both nitrification and denitrification (Zhu et al., 2013b) and to large soil $NO_3^-$ concentrations due to efficient cycling of deposited N in acid subtropical soils (Yu et al., 2016).

Chronically elevated rates of N deposition (30-65 kg $ha^{-1}$ $yr^{-1}$; Xu et al., 2015) have resulted in strong nutrient imbalances in southern Chinese forests, aggravating phosphorus (P) limitation (Du et al., 2016). Phosphorous deficiency in N-saturated forests restricts forest growth and thus



constrains its capability to retain N (Huang et al., 2015; Li et al., 2016), resulting in ample
amounts of mineral N ($NH_4^+$ and $NO_3^-$) being present in the soil solution. Accordingly, Hall &
Matson (1999) observed larger $N_2O$ emission in P-limited than in N-limited tropical forests after
1 year of repeated N addition. Likewise, previous N manipulation studies in forests of South
China reported pronounced stimulation of $N_2O$ emissions by N addition (Chen et al., 2016;
Wang et al., 2014; Zheng et al., 2016), supporting the idea that P limitation causes forests to be
more susceptible to N saturation and $N_2O$-N loss. In an N-limited tropical montane forest in
southern Ecuador, P addition alone (10 kg P $ha^{-1}$ $yr^{-1}$) had no effect on $N_2O$ emissions during the
first two years. However, $N_2O$ emission was smaller when P was added together with N (50 kg N
$ha^{-1}$ $yr^{-1}$) than treatments with N addition alone (Martinson et al., 2013). After continued
fertilization for three years, also P addition alone reduced $N_2O$ emissions at these sites (Müller et
al., 2015). In tropical China, with high N deposition (~ 36 kg $ha^{-1}$ $yr^{-1}$; Mo et al., 2008), P
addition (150 kg P $ha^{-1}$ $yr^{-1}$) to an old-growth forest revealed a similar pattern, with no initial
effect on $N_2O$ emissions (0-2 years) but a significant longer term effect (3 to 5 years) (Chen et al.,
2016; Zheng et al., 2016). In a secondary tropical forests in South China, Wang et al. (2014)
found no effect on $N_2O$ emissions of P alone (100 kg P $ha^{-1}$ $yr^{-1}$), and in treatments combining P
with N (100 kg N $ha^{-1}$ $yr^{-1}$),  $N_2O$ emissions even increased during the wet season. Meanwhile,
they observed a significant increase in soil microbial biomass after P addition, which is in line
with previous findings in tropical forest soils of South China (Liu et al., 2012). Thus, they
attributed the stimulating effect of P addition on $N_2O$ emissions to the larger nitrification and
denitrification potential of the increased soil microbial biomass. This was also proposed by Mori
et al. (2014), based on results from a short-term incubation study with P addition, excluding plant
roots.



As the sole biogenic sink for $CH_4$, upland soils play an important role in balancing terrestrial
$CH_4$ emissions (Ciais et al., 2013; Dutaur and Verchot, 2007). Atmospheric $CH_4$ uptake in soil is
mediated by the activity of methanotrophic bacteria, which oxidize $CH_4$ to $CO_2$ to gain energy
for growth. Well-drained forest and grassland soils are dominated by yet uncultured, high-
affinity methanotrophs residing in the upper soil layers (Le Mer and Roger, 2010). In addition to
edaphic factors (pH and nutrients), other parameters affecting the diffusion of $CH_4$ into the soil
(soil structure, moisture, temperature) are believed to be the major controllers for $CH_4$ uptake
(Smith et al., 2003). A number of studies have shown that excess N affects $CH_4$ fluxes in forest
soils (Liu and Greaver, 2009; Veldkamp et al., 2013; Zhang et al., 2008b). In general, N addition
promotes $CH_4$ uptake in N-limited soils by enhancing growth and activity of methanotrophs,
whereas excessive N input and N saturation inhibit $CH_4$ oxidation on an enzymatic level
(Aronson and Helliker, 2010; Bodelier and Laanbroek, 2004). P addition experiments in N-
enriched soils have shown positive effects on $CH_4$ uptake (Mori et al., 2013a; Zhang et al., 2011),
but the underlying mechanisms, i.e. whether P addition affects the methanotrophic community in
soils directly or alleviates the N-inhibition effect on $CH_4$ oxidation through enhanced N uptake
(Mori et al., 2013b; Veraart et al., 2015), remain unresolved.
Subtropical forests in South China show strong signs of N saturation, with exceedingly high
$NO_3^-$ concentrations in soil water (Larssen et al., 2011; Zhu et al., 2013b). Little is known about
how P addition affects N cycling and $N_2O$ emission in these acidic, nutrient-poor soils. Likewise,
the importance of increased mineral N concentrations for soil-atmosphere exchange of $CH_4$, and
how this is affected by P fertilization remain to be elucidated for soils of the subtropics. Here, we
assessed $N_2O$ and $CH_4$ fluxes in an N-saturated subtropical forest in SW China under ambient N
deposition and studied the effects of P addition on emission rates, nutrient availability and tree





growth. The objectives were i) to quantify ambient $N_2O$ and $CH_4$ emissions, ii) to test whether P
affects N cycling in a highly N-saturated forest and iii) to investigate the effect of P addition on
$N_2O$ and $CH_4$ emission.



## 2 Materials and Methods

### 2.1 Site description

The study site "TieShanPing" (TSP) is a 16.2 ha subtropical forest (29° 380 N, 106° 410 E; 450 m a.s.l.), about 25 km northeast of Chongqing, SW China. TSP is a naturally regenerated, secondary mixed coniferous-broadleaf forest, which developed after clear cutting in 1962 (Larssen et al., 2011). The forest stand is dominated by Masson pine (*Pinus massoniana*) and has a density of about 800 stems ha$^{-1}$ (Huang et al., 2015). Having a monsoonal climate, TSP has a mean annual precipitation of 1028 mm, and a mean annual temperature of 18.2 °C (Chen and Mulder, 2007). Most of precipitation (> 70%) occurs during the summer period (April to September). The soil is a loamy yellow mountain soil, classified as Haplic Acrisol (WRB 2014), with a thin O horizon (< 2 cm). In the O/A horizon, soil pH is around 3.7, and the mean C/N and N/P ratios are 17 and 16, respectively. In the AB horizon, which has a slightly higher pH, mean C/N is well above 20. More details on soil properties are presented in Table 1.

Annual N deposition at TSP measured in throughfall varies between 40 to 65 kg ha$^{-1}$ and is dominated by $NH_4^+$ (Yu et al., 2016). According to regional data, annual P deposition via throughfall is < 0.40 kg ha$^{-1}$ (Du et al., 2016). Strong soil acidification at TSP has resulted in severe decline in forest growth (Li et al., 2014; Wang et al., 2007), and in abundance and diversity of ground vegetation (Huang et al., 2015). Pronounced N saturation with strong $NO_3^-$ leaching from the top soil has aggravated P deficiency (Huang et al., 2015). The total P content in the O/A horizon is ~ 300 mg kg$^{-1}$, while $P_{Al}$ is smaller than 5 mg kg$^{-1}$ (Table 1).

### 2.2 Experimental Design



Three blocks, each having two 20 m * 20 m plots, were established near a hilltop on a gently
sloping hillside. A 5-m buffer strip separated the two plots in each block. In each block, plots
were assigned ad random to a reference (Ref) and a P treatment. On 4 May 2014, a single dose of
P fertilizer was applied as solid $NaH_2PO_4 \cdot 2H_2O$, at a rate of 79.5 kg P ha$^{-1}$. The amount of P
added was estimated from P adsorption isotherms (Supplementary Materials, Table S1 and
Figure S1), to ensure significantly increased available P in TSP soil. To apply P fertilizer evenly,
we divided each plot into a 5 m * 5 m grid and broadcasted the powdered fertilizer by hand in
each grid cell. The P dose applied at TSP was intermediate as compared to the 10 kg P ha$^{-1}$ yr$^{-1}$
applied by Müller et al. (2015) to a mountain forest in Ecuador and the 150 kg P ha$^{-1}$ yr$^{-1}$ applied
by Zheng et al. (2016) to a subtropical forest in South China.
Together with the addition of phosphate, the P-treated plots also received 59.0 kg ha$^{-1}$ of sodium
(Na). One month after the fertilizer application, Na$^+$ concentrations in soil water of the P
treatments were about 5 mg L$^{-1}$ at 5-cm depth and 3 mg L$^{-1}$ at 20-cm depth (Table S2). Although
somewhat larger than in the reference plots, the Na$^+$ concentration in soil water of the P
treatments are unlikely to have exerted a strong negative impact on plant and microbial activities.
**2.3 Sample collection and analyses**
Within each plot, triplicates of ceramic lysimeters (P80; Staatliche Porzellanmanufaktur, Berlin)
were installed at 5- and 20-cm soil depths in August 2013. To obtain water samples, 350-ml
glass bottles with rubber stoppers were pre-evacuated, using a paddle pump, and connected to the
lysimeters for overnight sampling. Between November 2013 and October 2015, we sampled soil
pore water bi-monthly in the winter season and monthly during the growing season. All water
samples were kept frozen during storage and transport. Concentrations of NH$_4^+$, NO$_3^-$, potassium





($K^+$), calcium ($Ca^{2+}$), and magnesium ($Mg^{2+}$) in soil water were measured at the Research Center
for Eco-Environmental Sciences (RCEES), Chinese Academy of Sciences, Beijing, using ion
chromatography (DX-120 for cations and DX-500 for anions).
In August 2013, soils from the O/A (0-3 cm), AB (3-8 cm) and B (8-20 cm) horizons were
sampled near the lysimeters for soil analysis. Total P and plant-available P contents were
monitored in samples collected from the O/A horizons every six months, starting two days
before P addition. Soil samples were kept cold (< 4 °C) during transport and storage. Before
analysis, soil samples were air dried and sieved (2 mm). Soil pH was measured in soil
suspensions (10 g dry soil and 50 ml deionized water) using a pH meter (PHB-4, Leici, China).
Total soil C and N contents were determined on dried and milled samples, using a LECO
elemental analyzer (TruSpec@CHN, USA). To measure total P, 1 g dry soil was digested with 5
ml of 6 M $H_2SO_4$ (Singh et al., 2005) and measured as ortho-phosphate by the molybdenum blue
method (Murphy and Riley, 1962). Ammonium lactate (0.01 M)-extractable P and $H_2O$-
extractable P ($P_{Al}$ and $P_{H2O}$, respectively) were measured as ortho-phosphate after extraction (1.5
g dry soil in 50 ml solution) (Singh et al., 2005). Ammonium oxalate (0.2 M)-extractable Fe, Al
and P were measured by inductive coupled plasma (7500; Agilent) after extraction (1.5 g dry soil
in 50 ml solution).
From August 2013 onwards, we measured $N_2O$ and $CH_4$ emissions in triplicate in micro-plots
close to the lysimeters, using static chambers (Zhu et al., 2013b). To investigate the immediate
effect of P addition on $N_2O$ emissions, we sampled the gas emissions once before (2 May) and
three times (7, 10 and 12 May) after the P application. Gas samples (20 ml) were taken 1, 5, 15
and 30 minutes after chamber deployment and injected into pre-evacuated glass vials (12 ml)
crimp-sealed with butyl septa (Chromacol, UK), maintaining overpressure to avoid





contamination during sample transport. Mixing ratios of $N_2O$, $CO_2$ and $CH_4$ were analyzed using
a gas chromatograph (Model 7890A, Agilent, US) at RCEES, equipped with an ECD for
detection of $N_2O$ (at 375 ºC with 25 ml min$^{-1}$ Ar/$CH_4$ as make up gas), a FID for $CH_4$ (250 ºC;
20 ml min$^{-1}$ $N_2$ as make-up gas) and a TCD for $CO_2$. Exchange rates between soil and
atmosphere (emission/uptake) were calculated from measured concentration change in the
chambers over time, applying linear or polynomial fits to the concentration data. Cumulative
$N_2O$ emissions over time were estimated by linear interpolation between measurement dates
(Zhu et al., 2013b).
From October 2013 onwards, litterfall was collected during the first week of every month in five
replicates per plot. Litterfall collectors were made of 1 m$^2$ nylon nets (1 mm mesh size), held in
place by four wooden poles 0.8 m above the ground. Fresh litter was dried at 65ºC. In early
November 2013 and 2014 (at the end of the growing season), we collected current-year pine
needles from several branches of three trees in each plot. The collected needles were dried at
65 ºC and the dry weight of 500 needles was determined. A subsample was dried at 80 ºC and
finely milled prior to chemical analysis at the Chinese Academy of Forestry. Total C and N were
measured using an elemental analyzer (FLASH 2000; Thermo Scientific; USA). The contents of
K, Ca, Mg and P in the needles were determined by ICP-AES (IRIS Intrepid II; Thermo
Scientific; USA) after digesting 0.25 g dry weight samples with 5 ml of ultra-pure nitric acid. In
November 2013, and 2014, and in February of 2015, we measured the height and the diameter at
breast height (DBH) of 6 to 10 Masson pines (only those with DBH > 5 cm) at each plot. These
data were used to estimate the standing biomass of Masson pines based on standard allometric
equations (Li et al., 2011; Zeng et al., 2008).



Daily average air temperature and sum of precipitation were monitored by a weather station
(WeatherHawk 232, USA) placed on the roof at the local forest bureau, in about 1 km distance
from the sampling site  (Yu et al., 2016).
**2.4 Statistical analyses**
Statistical analyses were performed with Minitab 16.2.2 (Minitab Inc., USA). All data were
tested for normality (Kolmogorov-Smirnov's test) and homoscedasticity (Levene's test) before
further analysis. If not normally distributed, the data were then normalized by logarithmic
transformation. Due to heterogeneity between blocks, data on gas fluxes and mineral N
concentrations are presented separately for each block. One-way ANOVA was used to evaluate
differences in gas fluxes, as well as nutrient concentrations in soil, soil water and plants between
treatments and blocks. Significance levels were set to $p < 0.05$, if not specified otherwise.



## 3 Results

### 3.1 Nutrient concentrations in soil and soil water

Addition of P resulted in a significant increase in soil P content in the O/A horizon, both as $P_{Al}$ and total P (Table 2). However, after 15 months, only $P_{AL}$ indicated an enhanced P status, while total soil P did not differ significantly from background values at the reference sites. P addition had no significant effect on soil pH, or soil C and N content. The $NO_3^-$ concentration in soil water collected at 5 cm depth varied seasonally, with significantly greater values (30-40 mg N $L^{-1}$) towards the start of the growing season in 2015 (April, Fig. S2), but not in 2014, likely due to dilution by abundant precipitation in February to March 2014. Addition of P resulted in significantly smaller $NO_3^-$ concentrations in soil water at 5 and 20 cm depth in blocks 2 and 3 but not in block 1 (Fig. 1). In general, the concentration of $NH_4^+$ in soil water was small (< 0.6 mg $L^{-1}$) and not affected by P addition (Fig. S3). At both depths, mean soil water concentrations of $Mg^{2+}$ and $Ca^{2+}$ were significantly smaller in the P-treated than the reference plots, and the sum of charge of base cations declined significantly in response to P addition (Fig. S4).

### 3.2 $N_2O$ and $CH_4$ fluxes: effects of P addition

During the experimental period, $N_2O$ fluxes varied seasonally (Fig. 2), showing a significant relationship with daily precipitation (Fig. S5a), but not with daily mean temperature (Fig. S4b). In the reference plots, mean $N_2O$ fluxes were generally below 50 μg N $m^{-2}$ $hr^{-1}$ in the dry, cool season, but reached up to 600 μg N $m^{-2}$ $hr^{-1}$ in the growing season (Fig. 2). Average and cumulative fluxes of $N_2O$ differed greatly between the three blocks (Figs. 3 and 4, respectively), with the greatest annual emission observed in the reference plot (7.9 kg N $ha^{-1}$) of block 2. Mean $N_2O$ fluxes during the 1.5 years after P addition were smaller in the P treatments than in the





references, the differences being significant in blocks 2 and 3 (Fig. 3). Cumulative $N_2O$
emissions showed that P addition resulted in a decrease in $N_2O$ emission by about 3 kg N ha$^{-1}$ yr$^{-1}$,
which is a 57% reduction on average (Fig. 4). No immediate effects (within days) of P addition
on $N_2O$ emission were observed (Fig. S6).
$CH_4$ fluxes varied greatly between blocks (Fig. 5).  Net-emission of $CH_4$ was observed in
summer 2013 (~ 80 µg C m$^{-2}$ hr$^{-1}$) in blocks 1 and 2, whereas block 3 showed $CH_4$ uptake. From
spring 2014 until October 2015, $CH_4$ fluxes were less variable in all blocks, with values
fluctuating around zero. A longer period of net-emission was observed in block 3 during the dry
season 2014. The fluxes did not correlate with either precipitation or air temperature (Fig.
S5c&d). In the 1.5 years following P addition, mean $CH_4$ fluxes indicated net $CH_4$ emission (~
+3.8 µg C m$^{-2}$ hr$^{-1}$) in the reference plots (except for block 1), whereas net $CH_4$ uptake (~ -6.5 µg
C m$^{-2}$ hr$^{-1}$) was observed in all P- treated plots (Fig. 6). The suppressing effect of P addition on
$CH_4$ emission was significant in blocks 2 and 3, similar to what was found for $NO_3^-$
concentrations and $N_2O$ fluxes.

**3.3 The effect of P addition on tree growth**

Throughout the 2-year experimental period, we observed no change in tree biomass (138 t ha$^{-1}$)
in response to P addition (Table S3). Likewise, there was no effect of P treatment on the 500-
needle weight (13 g on average). Between the two samplings in 2013 and 2014, we found
differences in chemical composition of the pine needles, but this effect was not linked to P
addition. Also, the C/N and N/P ratios of the needles (40 and 16, respectively) were hardly
affected by P addition. Monthly litterfall varied seasonally (Fig. S7), but no significant difference
was found between the reference and the P treated plots.





## 4 Discussion


Background $N_2O$ emission rates in the reference plots were relatively large, with mean values of
40 to 120 µg N m$^{-2}$ hr$^{-1}$ (Fig. 3). This is within the range previously reported for well-drained
hillslope soils at TSP (Zhu et al., 2013b), but greater than the rates reported for other forests in
South China. For instance, $N_2O$ emission rates averaged 37 µg N m$^{-2}$ hr$^{-1}$ in unmanaged sites at
Dinghushan (Fang et al., 2009; Tang et al., 2006) and up to 50 µg N m$^{-2}$ hr$^{-1}$ in N-fertilized sites
(Zhang et al., 2008a). TSP reference plots emitted on average 5.3 kg N ha$^{-1}$ yr$^{-1}$ (Fig. 4), which is
about 10% of the annual N deposition (50 kg ha$^{-1}$ yr$^{-1}$) (Huang et al., 2015). These fluxes were
well above average fluxes reported for tropical rainforests (Werner et al., 2007). High $N_2O$
emissions at TSP are likely due to the large N deposition rates (Huang et al. 2015), as suggested
by the similar trends indicated by data from a wide range of ecosystems (Liu et al., 2009). Also,
warm-humid conditions during monsoonal summers may stimulate $N_2O$ emissions (Ju et al.,
2011), as monsoonal rainstorms triggered peak fluxes (Fig. S5a) (Pan et al., 2003). The positive
correlation between precipitation and $N_2O$ emission peaks may indicate the importance of
denitrification as the dominant $N_2O$ source. This is supported by recent [15]N tracing experiments
at TSP (Zhu et al., 2013a; Yu et al., submitted).
Addition of P caused a significant decline in soil mineral N (predominantly $NO_3^-$) in two of three
blocks (Fig. 2), particularly during summers, when $NO_3^-$ concentrations were relatively high (Fig.
S2). At the same time, annual $N_2O$ emissions decreased by more than 50% (Figs. 3 and 4). These
findings are consistent with a number of previous studies (Baral et al., 2014; Hall and Matson,
1999; Mori et al., 2014). The reduction of $N_2O$ emissions in P treated soils was attributed to
decreased mineral N content, most likely due to stimulated plant uptake and/or microbial



assimilation. It is noteworthy, however, that there was no significant correlation between $N_2O$
emission rates and soil water $NO_3^-$ concentration in our study (Figs. 2 and S2), suggesting that
the suppressing effect of P on $N_2O$ emissions was indirect, probably by affecting the competition
for mineral N between plant roots and microbes (Zhu et al., 2016). In contrast to our study, P-
addition experiments in South Ecuador (Martinson et al., 2013) and South China (at Dinghushan
Biosphere Reserve (DHSBR); Zheng et al., 2016) found no effect of a single P addition on $N_2O$
emission during the first two years after application. However, significant reduction in $N_2O$
emission was observed after three to five years with continuous P addition, both at the
Ecuadorian and the Chinese site (Chen et al., 2016; Müller et al., 2015). For the montane forest
site in Ecuador, the observed delay in $N_2O$ emission response to P addition may be explained by
the moderate amount of P added (10 kg P ha$^{-1}$ yr$^{-1}$; Martinson et al., 2013). Moreover, the
experiments were conducted in a forest with low ambient N deposition (~ 10 kg N ha$^{-1}$ yr$^{-1}$) and
$N_2O$ fluxes (~ 0.36 kg N ha$^{-1}$ yr$^{-1}$ in the reference plot) (Martinson et al., 2013; Müller et al.,
2015). By contrast, the DHSBR site in South China receives 36 kg of atmogenic N ha$^{-1}$ yr$^{-1}$,
which is only slightly smaller than the N deposition at our site (Huang et al., 2015), and showed
larger $N_2O$ emission rate than the Ecuadorian site (~ 0.88 kg N ha$^{-1}$ yr$^{-1}$ in the reference plot;
Zheng et al., 2016). However, forests do not always display a straightforward relationship
between N deposition and $N_2O$ emissions. Manipulation experiments in the European NITREX
project, for instance, revealed a much stronger correlation of $N_2O$ emissions with soil $NO_3^-$
leaching than with N deposition (Gundersen et al., 2012). Indeed, KCl-extractable mineral N at
the DHSBR site (~ 40 mg kg$^{-1}$; Zheng et al., 2016) were several-fold smaller than at our site (>
100 mg kg$^{-1}$; Zhu et al., 2013b), indicating that DHSBR is less N-saturated than TSP. This
suggests that the response of $N_2O$ emission to P addition might depend on the N status of the soil.



The fact that numerous studies found apparent suppression of $N_2O$ emission in short-term
experiments (< 2 years) in N + P treatments, but not in treatments with P alone, supports this
idea (Müller et al., 2015; Zhang et al., 2014b; Zheng et al., 2016).
Other studies have observed increased $N_2O$ emissions upon P addition (Mori et al., 2013c; Wang
et al., 2014). In an *Acacia mangium* plantation, fertilized with P, Mori et al. (2013b&c) found
that $N_2O$ emissions were stimulated in the short-term but reduced in the long-term. While
suppression of $N_2O$ emission by P has been attributed to increased plant N uptake (Mori et al.,
2014), increased $N_2O$ emission are generally explained by enhanced microbial biomass (Liu et
al., 2012) and denitrification activities (Ehlers et al., 2010; He and Dijkstra, 2015). $N_2O$
emissions measured frequently after P addition at our site in May 2014 were not different from
fluxes in untreated reference plots (Fig. S5). This may indicate that plant uptake at TSP is more
important for the effect of P addition on $N_2O$ emissions than changes in microbial activity, which
are expected to occur more rapidly.
Two of three reference plots at TSP showed net $CH_4$ emission for extended periods of the year
(Figs. 5 and 6). Also, long-term $CH_4$ fluxes sampled between 2012 and 2014 on TSP hillslopes
near-by (Fig. S8; Zhu et al., unpublished data) showed net $CH_4$ emission. This is in contrast to
the generally reported $CH_4$ sink function of forested upland soils (Ciais et al., 2013; Dutaur and
Verchot, 2007). For example, $CH_4$ uptake rates reported for South Chinese forest soils range
from 30 to 60 µg C $m^{-2}$ $hr^{-1}$ (Fang et al., 2009; Tang et al., 2006; Zhang et al., 2014a). As $CH_4$
fluxes at our sites were not correlated with climatic factors (Fig. S5c and d), $CH_4$ emissions
cannot be explained by transiently wet conditions. One reason for the net-$CH_4$ emission observed
at TSP could be inhibition of $CH_4$ oxidation activity by $NH_4^+$, as reported previously (Bodelier
and Laanbroek, 2004; Zhang et al., 2014a). The concentration of $NH_4^+$ in the soil water was



rather small (< 0.5 g L$^{-1}$ ; Fig. S3), which does not preclude, however, that NH$_4^+$ availability from
the soil exchangeable pool is high. Zhu et al. (2013b) found extraordinarily high KCL-
extractable NH$_4^+$ in TSP surface soils, likely reflecting the large atmogenic NH$_4^+$ input at our site
(Huang et al., 2015).
P addition had a significant impact on CH$_4$ fluxes, changing the soil from a net source to a net
sink on an annual basis (Fig. 6). However, the uptake rates of CH$_4$ in the P treatments remained
smaller than those reported for forest soils in tropical China (Tang et al., 2006; Zhang et al.,
2008b). The stimulating effect of P addition on CH$_4$ uptake is consistent with previous studies
(Mori et al., 2013a, 2013b; Zhang et al., 2011), and has been attributed to lessening the NH$_4^+$
inhibition of methane oxidation. Unfortunately, we did not measure KCl-extractable NH$_4^+$ in our
study, but a decline of available NH$_4^+$, which is the substrate for nitrification, is likely as NO$_3^-$
concentrations in soil water were significantly smaller with in the P-treatments (Fig. 2). P
addition may also result in a change of the taxonomic composition of the methane oxidizing
community (Mori et al., 2013a; Veraart et al., 2015). Alternatively, CH$_4$ oxidation may be
stimulated by increased CH$_4$ diffusion into the soil, due to enhanced root growth and increased
transpiration in P-amended plots (Zhang et al., 2011). Given the high degree of N saturation of
TSP forest (Huang et al., 2015), it is likely that the reason for the observed reduction in CH$_4$
emissions in response to P fertilization was due to alleviating the NH$_4^+$ inhibition of the methane
monooxygenase enzyme (Veldkamp et al., 2013), rather than a direct P-stimulation of
methanotrophic activity (Veraart et al., 2015).
Shortly after fertilizer application, we observed a modest, albeit significant increase of Na$^+$
concentration in soil water (Table S2). Other studies have documented the toxicity of excess Na$^+$
in soil water to plant and microbial activities (Rengasamy et al., 2003; Wong et al., 2008).



However, the occurrence of $Na^+$ toxicity at the treated plots, affecting N turnover processes, is
unlikely, as $Na^+$ concentrations in soil water, within one month after application (Table S2), did
not exceed 5 mg $L^{-1}$, far below the values commonly assumed to indicate the toxicity threshold
(40 to 100 mg $L^{-1}$) (Bernstein 1975). The frequent precipitation in the humid forest of this study
(Yu et al., 2016), both prior and following the addition of $NaH_2PO_4.2H_2O$ (Fig. 2), efficiently
diluted and leached $Na^+$, thus minimizing toxic effects.
P application significantly increased plant-available P in the P-limited TSP soil (Table 2).
Meanwhile, concentrations of leachable base cations ($K^+$, $Mg^{2+}$, $Ca^{2+}$) in soil water decreased
(Fig. S4), as expected from the reduction of $NO_3^-$ concentrations in the P-treatments (Mochoge
and Beese, 1986). We observed no sign of stimulated forest growth or increased N uptake by
plants within the relatively short period of our study (Table S3 and Fig. S7), which makes it
difficult to link the observed reduction in mineral N in the soil solution to plant growth (Fig. 2).
When interpreting the observed P effect on $NO_3^-$ concentrations in soil water, several aspects
need to be considered. Firstly, two years of observation may be too short to detect any significant
$NO_3^-$ uptake by plants, given the commonly large variabilities in tree biomass estimates
(Alvarez-Clare et al., 2013; Huang et al., 2015). Secondly, a significant proportion of the added
P, and of excess N, may have been assimilated by the understory biomass, which was not
assessed in this study. Previously, understory vegetation has been reported to quickly respond to
P addition (Fraterrigo et al., 2011). Thirdly, as long-term N saturation and acidification at TSP
has reduced the forest health (Lu et al., 2010; Wang et al., 2007), we may not expect immediate
response of forest growth to P addition. Large needle N/P ratios (17-22, Table S3) indicated that
P limitation for tree growth was not relieved 1.5 years after P addition (Li et al., 2016). Therefore,





enhanced N uptake by understory growth and/or soil microbial biomass may have been the main
mechanisms responsible for observed $NO_3^-$ decline in the P-treated soil (Hall & Matson 1999).
Overall, our study demonstrates that chronically high N deposition has transformed TSP soils to
a regional hotspot for $N_2O$ and $CH_4$ emission. Within the short experimental period of 1.5 years,
P fertilization was shown to significantly decrease $NO_3^-$ concentrations in soil water and to
reduce both $N_2O$ and $CH_4$ emissions. These findings provide a promising starting point for
improving forest management towards GHG abatement targets, taking into account the P and N
status of subtropical soils in the region.





## 5 Acknowledgement

Longfei Yu thanks the China Scholarship Council (CSC) for supporting his PhD study. Support from the Norwegian Research Council to project 209696/E10 'Forest in South China: an important sink for reactive nitrogen and a regional hotspot for $N_2O$?' is gratefully acknowledged. We thank Prof. Wang Yanhui, Prof. Duan Lei, Dr. Wang Zhangwei, Zhang Yi, Zhang Ting, Zou Mingquan for their help during sample collection and data analysis. Dr. Zhu Jing is gratefully acknowledged for unpublished data on long-term $CH_4$ fluxes in the TSP catchment.




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





**Table 1** Background soil properties of the experimental plots at Tieshanping (TSP). Values are

| | Soil Layer | pH | Total C | Total N | Total P | C/N | N/P |
|---|---|---|---|---|---|---|---|
| | | | g kg⁻¹ | g kg⁻¹ | mg kg⁻¹ | | |
| | O/A (0-3 cm) | 3.7 (0.1) | 80.7 (32.3) | 4.8 (1.7) | 308 (57) | 17.0 (2.5) | 15.5 (5.7) |
| Block 1 | AB (3-8 cm) | 3.8 (0.0) | 23.9 (9.3) | 1.3 (0.6) | -* | 20.0 (3.0) | - |
| | B (8-20 cm) | 3.9 (0.2) | 8.6 (1.2) | < 0.05 | - | - | - |
| | O/A (0-3 cm) | 3.6 (0.1) | 77.6 (13.4) | 4.7 (0.8) | 297 (44) | 16.7 (1.3) | 15.7 (2.8) |
| Block 2 | AB (3-8 cm) | 3.7 (0.1) | 20.2 (5.3) | 1.0 (0.3) | - | 21.4 (3.3) | - |
| | B (8-20 cm) | 3.9 (0.1) | 7.1 (1.6) | < 0.05 | - | - | - |
| | O/A (0-3 cm) | 3.6 (0.1) | 67.0 (15.5) | 3.8 (0.8) | 223 (45) | 17.4 (0.6) | 17.2 (3.7) |
| Block 3 | AB (3-8 cm) | 3.6 (0.1) | 21.0 (7.9) | 1.1 (0.5) | - | 24.5 (4.6) | - |
| | B (8-20 cm) | 3.8 (0.1) | 7.2 (1.5) | < 0.05 | - | - | - |
| | Soil Layer | $P_{H2O}$ | $P_{Al}$ | $Al_{ox}$ | $Fe_{ox}$ | $P_{ox}$ | $P_{ox}$ / |
| | | mg kg⁻¹ | mg kg⁻¹ | mg kg⁻¹ | mg kg⁻¹ | mg kg⁻¹ | ( $Al_{ox}$ + $Fe_{ox}$ ) |
| | O/A (0-3 cm) | < 5.0 | 5.8 (1.4) | 1700 (513) | 1933 (350) | 85.8 (22.6) | 0.025 (0.008) |
| Block 1 | AB (3-8 cm) | < 5.0 | 2.1 (0.6) | 1217 (243) | 1692 (493) | 47.1 (22.0) | 0.016 (0.007) |
| | B (8-20 cm) | < 5.0 | < 1.0 | 1083 (90) | 1158 (249) | 29.3 (28.6) | 0.012 (0.011) |
| | O/A (0-3 cm) | < 5.0 | 5.9 (1.0) | 1500 (238) | 1792 (215) | 79.2 (21.5) | 0.024 (0.007) |
| Block 2 | AB (3-8 cm) | < 5.0 | 1.6 (0.4) | 925 (149) | 1517 (320) | 37.2 (10.7) | 0.016 (0.006) |
| | B (8-20 cm) | < 5.0 | < 1.0 | 892 (209) | 1033 (413) | 16.1 (10.5) | 0.009 (0.007) |
| | O/A (0-3 cm) | < 5.0 | 4.1 (0.9) | 1367 (180) | 1667 (168) | 50.7 (10.9) | 0.017 (0.003) |
| Block 3 | AB (3-8 cm) | < 5.0 | 4.4 (4.0) | 1075 (128) | 1350 (150) | 24.8 (8.3) | 0.010 (0.002) |
| | B (8-20 cm) | < 5.0 | < 1.0 | 992 (130) | 875 (138) | 8.0 (2.0) | 0.004 (0.001) |

means and standard deviations in parenthesis (n = 6)[†].

$P_{H2O}$ = Water extractable P, $P_{Al}$ = Ammonium extractable P,
$Al_{ox}$ = Oxalate extractable Al, $Fe_{ox}$ = Oxalate extractable Fe, $P_{ox}$ = Oxalate extractable P.
[†] Soils were sampled in August 2013.
[*] Data not available



**Table 2** Soil pH, C, N and P contents in the O/A horizon (0-3 cm) in Reference and phosphate (P)

| | | pH | Total C | Total N | C/N | $P_{Al}$ | Total P |
|---|---|---|---|---|---|---|---|
| | | | g kg[-1] | g kg[-1] | | mg kg[-1] | mg kg[-1] |
| 13/08/02 | Ref | 3.7 (0.1)[bc†] | 8.3 (2.3)[ab] | 0.5 (0.1)[bcd] | 16.9 (1.1)[bcd] | 5.4 (1.4)[c] | 292 (46)[bc] |
| | P | 3.6 (0.1)[c] | 6.7 (2.0)[b] | 0.4 (0.1)[bd] | 17.1 (2.1)[bc] | 5.1 (1.3)[c] | 260 (70)[c] |
| 14/05/02 | Ref | 3.7 (0.1)[abc] | 12.2 (4.2)[a] | 0.9 (0.3)[a] | 13.7 (1.5)[e] | 19.0 (8.0)[c] | 336 (65)[bc] |
| | P | 3.8 (0.2)[abc] | 9.0 (3.5)[ab] | 0.7 (0.2)[abc] | 14.2 (2.8)[de] | 13.7 (5.2)[c] | 270 (72)[bc] |
| 14/05/10 | Ref | 3.8 (0.1)[abc] | 9.9 (2.1)[ab] | 0.7 (0.2)[ab] | 14.0 (0.7)[e] | 15.4 (7.0)[c] | 304 (49)[bc] |
| | P | 3.9 (0.3)[ab] | 8.0 (1.9)[ab] | 0.6 (0.1)[bcd] | 14.3 (1.3)[cde] | 174 (114)[a] | 572 (242)[a] |
| 14/12/02 | Ref | 3.8 (0.1)[abc] | 10.5 (3.6)[ab] | 0.7 (0.3)[ab] | 14.5 (1.3)[cde] | 14.2 (7.4)[c] | 328 (102)[bc] |
| | P | 3.9 (0.2)[abc] | 9.5 (2.1)[ab] | 0.7 (0.1)[abc] | 14.0 (0.8)[e] | 66 (24)[ab] | 442 (106)[ab] |
| 15/08/02 | Ref | 3.9 (0.2)[ab] | 8.3 (2.2)[ab] | 0.4 (0.1)[cd] | 20.5 (2.5)[a] | 13.4 (6.2)[c] | 291 (61)[bc] |
| | P | 4.0 (0.2)[a] | 6.5 (1.9)[b] | 0.3 (0.1)[d] | 19.7 (2.2)[ab] | 57 (36)[ab] | 383 (136)[bc] |

treatments. Values are means and standard deviations in parenthesis (n = 9).

[θ] P addition was conducted on 14/05/04, after the first two sampling dates.
[†] Different letters indicate significance in difference.




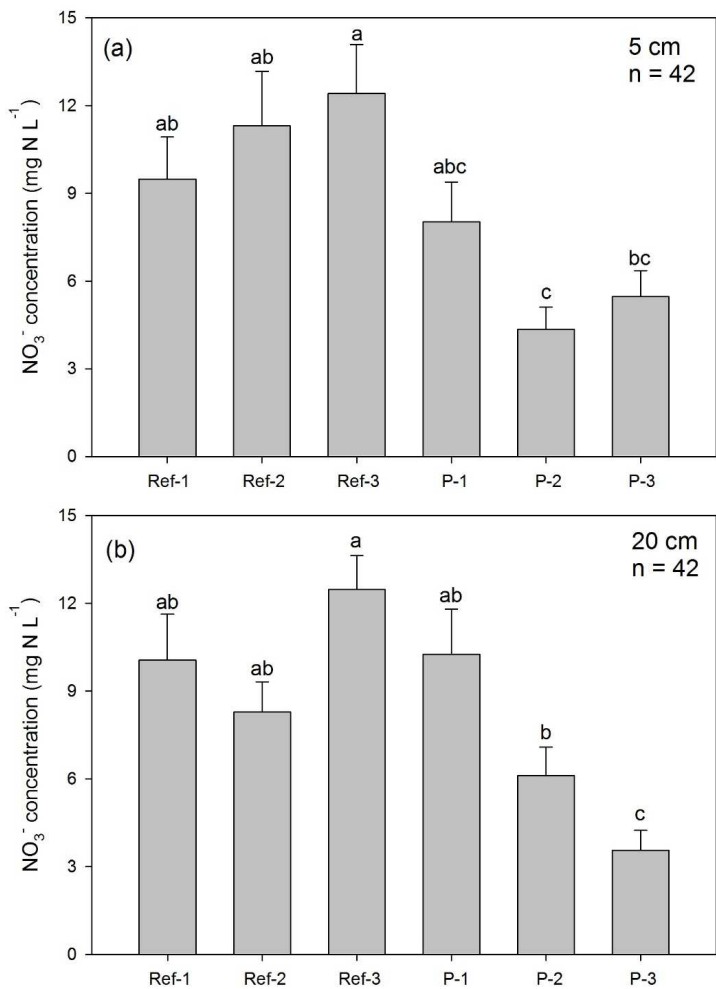

**Fig. 1** Mean $NO_3^-$ concentrations in soil water at 5 (a) and 20 (b) cm depths from three blocks

with Reference and P treatment, 1.5 years after P addition; different letters indicate significant

differences between the treatments and blocks

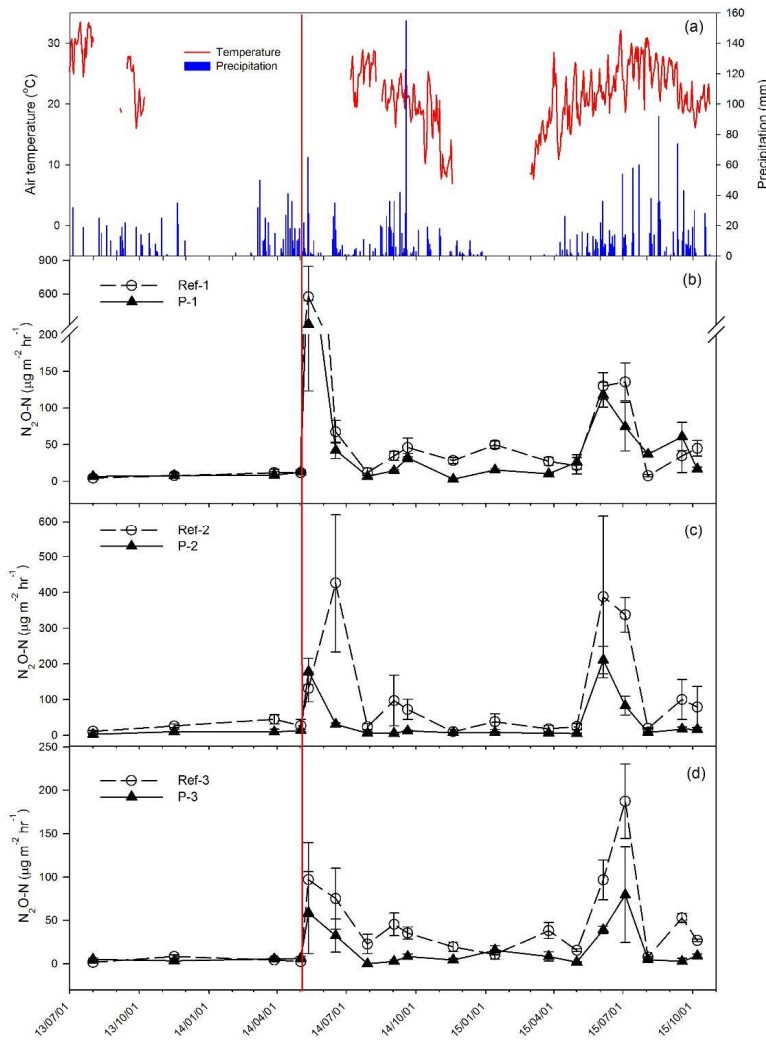


**Fig. 2** Daily mean air temperature and precipitation (a), and monthly mean $N_2O$ fluxes in

Reference and P treatments in each of the three blocks (b-d); the red line gives the date of P

addition.





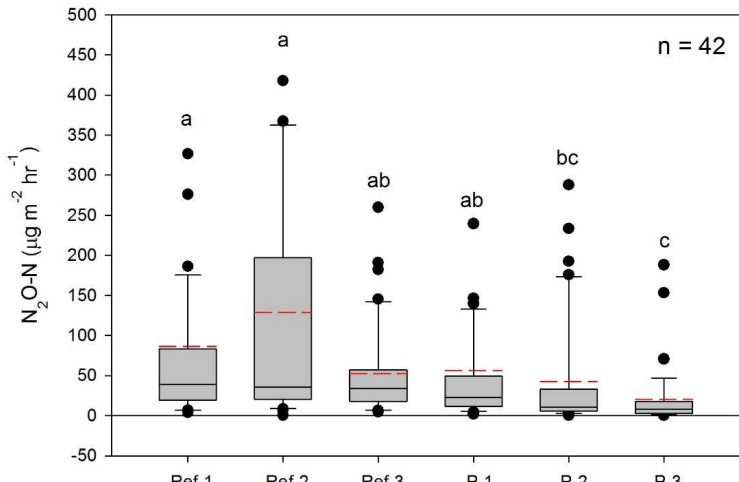


**Fig. 3** Box whisker plots for $N_2O$ fluxes in three blocks with Reference and P treatments
throughout 1.5 years after the P addition; red dash lines indicate mean values; different letters
indicate significant differences between treatments and blocks.





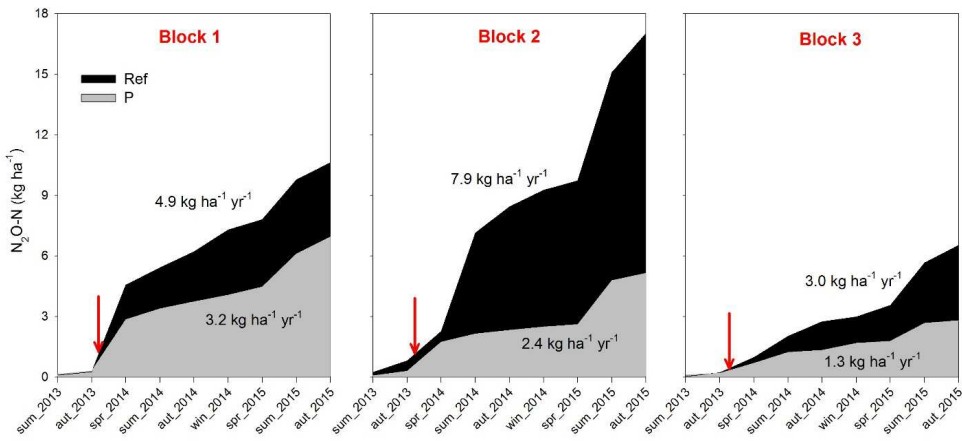


**Fig. 4** Cumulative N$_2$O emissions for three blocks with Reference and P treatments during two
years; the red arrows refer to the date when P addition was conducted.







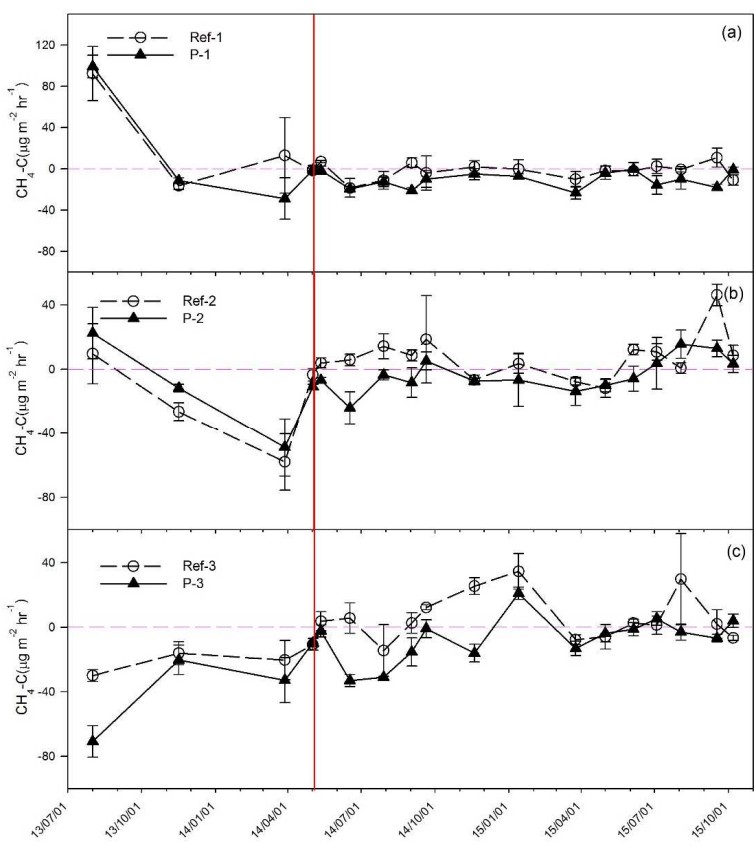


**Fig. 5** Monthly mean CH$_4$ fluxes for three blocks (a-c) with Reference and P treatments during

two years; the horizontal broken line indicates zero flux the red line refers to the date of P

addition.





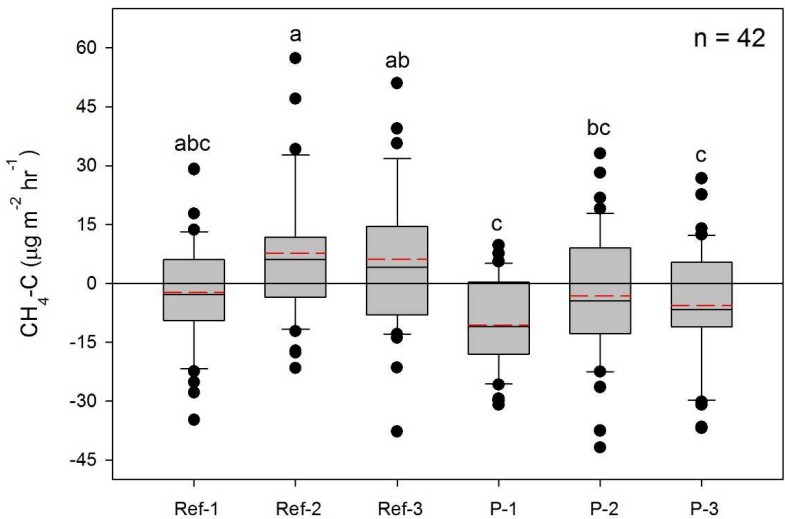


**Fig. 6** Box whisker plots of $CH_4$ fluxes for three blocks with Reference and P treatments 1.5

years after the P addition; red dash lines indicate mean values; the small letters indicate the
significance levels among the treatments and blocks.