# Peer review of "Phosphorus addition mitigates N2O and CH4 emissions in N saturated subtropical forest, SW China"

_Biogeosciences, 2016_

## Referee Comment (RC1) · Anonymous Referee #1 · 30 Nov 2016

General comments:

Yu and co-authors present an interesting dataset of nitrous oxide (N2O) and methane (CH4) flux, nitrate (NO3-) concentration and other ancillary measurements from a forest ecosystem P-addition experiment conducted in a N-saturated (N-deposition is about 40 to 65 kg ha-1 yr-1) secondary Masson pine-dominated forest at TieShanPing (TSP), Chongqing, SW China (developed after a clear cut about 50 years ago) over a period of 18 months. After a single dose of P (applied as solid NaH2PO4*2H2O at a rate of 79.5 kg P ha-1) was added, Yu and co-authors found out that both N2O and CH4 emissions and NO3- concentrations in soil water decreased following P-addition during the 18-months period. They speculate that P-addition may have stimulated mineral N uptake by P-limited plants or microbes leading to decreased NO3- concentrations and N2O emissions. Concomitantly, decreasing mineral N concentrations may have relieved N-

inhibition of microbial CH4 oxidation leading to decreased CH4 emissions. Spatial and temporal dynamics of nutrient imbalances and their effects on biogeochemistry in forest ecosystems are very complex and difficult to decipher but important to understand in order to predict impacts of global change processes on trace gas fluxes in forest ecosystems. The results from Yu and co-authors are very interesting and valuable. Nutrient-addition experiments in whole forest ecosystems are very difficult to conduct but their results are often much more realistic than countless laboratory experiments trying to mimic whole ecosystem conditions.

However, I have major concerns and recommend a reanalysis of the dataset:

1) I would suggest to reanalyze the whole dataset using linear mixed effects models (Koehler et al. 2009 and Jones et al. 2016) to account for repeated measurements (monthly measurements over 1.5 years) and for within-group variance of a stratification (block-design of the study) which has not been done so far.

2) Please take into account that all of your replicates in your plot are pseudo-replicates because they depend on one single block (which is your true replicate). In total you have three independent samples for the P-addition plots and three independent reference samples. Consequently, the dataset and figures should be reanalyzed and presented appropriately. If the blocks are so heterogenous you may show patterns in different blocks or outliers but your whole discussion should focus on significant results of the reanalyzed dataset based on plot means and not subplots.

3) I doubt that chronic N deposition alone has transformed TSP soils to a regional hotspot for N2O and CH4 emissions. You have not measured that. Changes in soil bulk density/soil compaction following a clear-cut about 50 years ago may be even more important. Water-filled pore space or gravimetric water content in soils are major controllers of N2O and CH4 production. These variables are almost always measured in soil trace gas flux studies and highly depend on soil bulk density which has also not been measured or is not presented in the manuscript. Since there was a clear-cut
at the TSP site about 50 years ago and the soil type at this site is a Haplic Acrisol, where clay translocation processes in mineral horizons form clay-enriched horizons. Such a soil is prone to soil compaction. Especially in these clay-enriched horizons soil compaction may lead to increased bulk densities, may promote oxygen limitation and therefore increase rates of microbial denitrification and methanogenesis that eventually may lead to net $CH_4$ and $N_2O$ emissions from this forest site. It would be great if authors could provide data about soil bulk densities, soil properties in general and soil water content variables.

Minor comments: Line 28 to 30: Nutrient imbalances in forest ecosystems and their effects on greenhouse gas emissions are very complex and shift in space and time. The present study analyzed effects of a single dose of P fertilizer on trace gas fluxes in an approximately 50-years old secondary forest over a period of 18 moths. It is simply too daring to extrapolate results from this special forest site to acid forest soils in general. Please do not speculate so much.

Line 127: change to: "In each block, plots were randomly assigned to a reference (Ref) and a P treatment."

Line 164-165: How often and when did you measure trace gas fluxes? Did you take water samples ($NH_4+$ and $NO_3-$ concentrations) at the same time? A different sampling time may explain the lack of correlation between $N_2O$ emissions and $NO_3-$ concentrations.

Line 251-254: Why do you use different units ($\mu$g N m-2 hr-1 ; kg N ha-1 yr-1) for the same variable?

Line 282: change to atmospheric

Line 362: Your study does not demonstrate that chronically high N deposition has transformed TSP soil to a regional hotspot for $N_2O$ and $CH_4$ emissions. You have not measured that.

Figures: Please provide information about sample size and if you use SD or SE in all of your figures.

Reference: Jones et al. (2016) Biogeosciences, 13, 4151–4165 Koehler et al. (2009) Global Change Biology, 15, 2049 -2066

―――――――――――――――――

---

## Referee Comment (RC2) · Anonymous Referee #2 · 4 Dec 2016

General comments The authors reported en experiment of P addition (79 kg P ha-1 yr-1, applied as NaH2PO4 powder) to an N saturated, Masson pine-dominated forest at TieShanPing (TSP), Chongqing, SW China for a period of 18 months. In the experiment, they measured soil fluxes of N2O and CH4, soil chemistry and plant growth. They found that P addition significantly decreased soil N2O emissions and turned the soil from CH4 emissions into a net sink. The experiment is appropriate. Data interpretation was logical and supported their conclusion. The study is for a type of ecosystem (subtropical, high ambient N deposition, N-saturated forest soil) for which such information is lacking. Their findings are interesting, can help us understand the interaction effect of N and P on greenhouse gas emission and also have implication for the forest management (such as P fertilization). In addition, the manuscript is also well organized and well written. I have only some minor concerns which I would like to discuss with

the authors or maybe helpful for improving the manuscript, please see details below. Specific comments 1) P7, Line117. "annual N deposition at TSP measured in throughfall varies between 40-65 kg ha-1". What is the N deposition in precipitation there? Because N deposition in throughfall may be affected by difference in species, structure etc. of the forest, N deposition in precipitation is better data for comparing different forests. 2) P9, L165. Please give the locations (in the center of the plot?) where you measured the gas emission in the plots. 3) P9, L165-167. "to investigate the immediate effect of P addition on …..... (7, 10 and 12 May) after the P application." Did you show these results in the result or discussion section? If not, please delete it. 4) P11, Line 195 Statistical analyses. In the Experimental Design, the author showed that three blocks were established and two plots in each block in the study forest. In each block, plots were assigned randomly to a reference (Ref) and a P treatment. Did you try One-way Repeated-Measures ANOVA to exam the treatment effect for the emission of CH4 and N2O, due to measuring the gas repeatedly. 5) P19, L362-363. "Overall, our study demonstrates that chronically high N deposition has transformed TSP soils to a regional hotspot for N2O and CH4 emission. " It is not clear for me. Could you explain it? 6) P25, L565-568, "Zhang et al., 2014. Responses of nitrous oxide emissions to nitrogen and phosphorus …....." has been published in Biogeosciences, please replace Biogeosciences Discuss. 7) P27, L585, Table 1. How did you get n=6? Did you mixed samples in each plot? 8) P28, L591, Table 2. The yearly variation for some data is big. For example, PAL in the ref plots was 5.4 in Aug. 2013, but was 13.4 in Aug. 2015. Do you have any explanation for it?

---

## Referee Comment (RC3) · Anonymous Referee #3 · 5 Dec 2016

This paper report the effects of P addition on leachate chemistry and gas exchange of N2O and CH4 in a high N deposition forest of the warm and humid part of China. The results are interesting and the paper easy to read. However, the statistical treatment is not ideal, since the repeated nature of the measurements seems not to have been considered in the model tests. I see this problem is well addressed by the other reviewers. This needs to be addressed although it will not change the outcome and major conclusions. Below is my mainly minor comments listed by line number 70: 'and' = 'but' 71: delete: 'even' 123: spell out what means PAI here at first appearance. 126: 5-m buffer is a bit narrow, but cannot be changed 127: 'ad' = 'at' 130: '. . .. in the TSP soil.' 141: 'Within each plot, three ceramic . . ..' 145: 'winter' = 'dormant' or 'dormant and dry' ?  200: should be repeated ANOVA of some kind.  It seems to me that the statistical analysis is not optimal. 206: check the subscript on P 290: I am not in favor

of discussing degrees of N-saturation; I would instead say 'DHSRB is less N-rich with lower inorganic N availability than TSP' 303: 'frequently' = 'shortly' 306: delete 'TSP' here, implied in nearby 312-318: inhibition by NH4 cannot explain emission only lower uptake rates; so delete or reformulate 330: like line 290; reformulate 328-334: I would suggest that both reason (and others as well) may have contributed 351-353: I do not understand this; what is the 'tree biomass estimates' doing here? 353-356: Why not, this should be simple and not much effort? 436-38: something wrong in this ref 587: add 'lactate' 589, 594+595 add these lines to the table legend.
* * *

---

## Author Comment (AC1) · 26 Jan 2017

Dear Editor,

Please find below our response to the referees' comments on the manuscript "Phosphorus addition mitigates N2O and CH4 emissions in N-saturated subtropical forest, SW Chna". We would like to thank anonymous referees for their valuable and detailed comments. The answers are presented in bold and red fonts.

We also thank the editor for handling our manuscript. We remain at your disposal for any further enquiries.

Sincerely, Longfei Yu

[Figure]

Please also note the supplement to this comment:
http://www.biogeosciences-discuss.net/bg-2016-470/bg-2016-470-AC1-supplement.pdf

[Figure]

**Supplement:**

Received and published in BG Discussion on 03 Nov. 2016.

**Referee #1**

General comments:

Yu and co-authors present an interesting dataset of nitrous oxide (N2O) and methane (CH4) flux, nitrate (NO3-) concentration and other ancillary measurements from a forest ecosystem P-addition experiment conducted in a N-saturated (N-deposition is about 40 to 65 kg ha-1 yr-1) secondary Masson pine-dominated forest at TieShanPing (TSP), Chongqing, SW China (developed after a clear cut about 50 years ago) over a period of 18 months. After a single dose of P (applied as solid NaH2PO4*2H2O at a rate of 79.5 kg P ha-1) was added, Yu and co-authors found out that both N2O and CH4 emissions and NO3- concentrations in soil water decreased following P-addition during the 18-months period. They speculate that P-addition may have stimulated mineral N uptake by P-limited plants or microbes leading to decreased NO3- concentrations and N2O emissions. Concomitantly, decreasing mineral N concentrations may have relieved N inhibition of microbial CH4 oxidation leading to decreased CH4 emissions. Spatial and temporal dynamics of nutrient imbalances and their effects on biogeochemistry in forest ecosystems are very complex and difficult to decipher but important to understand in order to predict impacts of global change processes on trace gas fluxes in forest ecosystems. The results from Yu and co-authors are very interesting and valuable. Nutrient-addition experiments in whole forest ecosystems are very difficult to conduct but their results are often much more realistic than countless laboratory experiments trying to mimic whole ecosystem conditions.

**R 1.0: We thank referee #1 for the appraisal and valuable comments intended to improve the presentation of the data. We have responded below to the comments.**

However, I have major concerns and recommend a reanalysis of the dataset:

1) I would suggest to reanalyze the whole dataset using linear mixed effects models (Koehler et al. 2009 and Jones et al. 2016) to account for repeated measurements (monthly measurements over 1.5 years) and for within-group variance of a stratification (block-design of the study) which has not been done so far.

**R 1.1: We agree with the referee that Linear Mixed Effects Models would be better to capture within-group variances, even though the overall result may not differ substantially. In the revision, we will document the details of our statistical tests for different parameters, and present our final outcomes with reanalyzed results.**

2) Please take into account that all of your replicates in your plot are pseudo-replicates because they depend on one single block (which is your true replicate). In total you have three independent samples for the P-addition plots and three independent reference samples. Consequently, the dataset and figures should be reanalyzed and presented appropriately. If the blocks are so heterogenous you may show patterns in different blocks or outliers but your whole discussion should focus on significant results of the reanalyzed dataset based on plot means and not subplots.

**R 1.2: We agree with the referee's suggestions and will change the manuscript accordingly.**

3) I doubt that chronic N deposition alone has transformed TSP soils to a regional hotspot for N2O and CH4 emissions. You have not measured that. Changes in soil bulk density/soil compaction following a clear-cut about 50 years ago may be even more important. Water-filled pore space or gravimetric water content in soils are major controllers of N2O and CH4 production. These variables are almost always measured in soil trace gas flux studies and highly depend on soil bulk density which has also not been measured or is not presented in the manuscript. Since there was a clear-cut at the TSP site about 50 years ago and the soil type at this site is a Haplic Acrisol, where clay translocation processes in mineral horizons form clay-enriched horizons. Such a soil is prone to soil compaction. Especially in these clay-enriched horizons soil compaction may lead to increased bulk densities, may promote oxygen limitation and therefore increase rates of microbial denitrification and methanogenesis that eventually may lead to net CH4 and N2O emissions from this forest site. It would be great if authors could provide data about soil bulk densities, soil properties in general and soil water content variables.

**R 1.3: N saturation, indicated by significant nitrate leaching from soils, has been previously confirmed at TSP, (Chen and Mulder, 2007; Huang et al., 2015; Zhu et al., 2013b). Accordingly, annual $N_2O$ fluxes previously observed at TSP were among the highest fluxes reported in literature (Zhu et al., 2013b). Chronic N deposition does not only increase soil N availability, but also contributes to soil acidification (Huang et al., 2015), which is believed to facilitate large $N_2O$ emission factors ($N_2O$ flux: total N input) (Liu et al., 2014). Therefore, based on the previous findings, our study mainly addresses the P-addition effect on GHG emissions at TSP.**

**We also agree with the referee that edaphic factors, specific to Haplic Acrisols, play a role in the $N_2O$ and $CH_4$ emissions in TSP. Though WFPS was not directly monitored throughout our observation, the range of WFPS and its effect on $N_2O$ emissions in TSP have been well presented and studied by Zhu et al (2013b). The clear cut of the forests in the 1950s may affect the edaphic conditions in clay-rich horizon in soil, but not likely for the organic top horizon, where microbial activities such as denitrification were found to be strongest (Zhu et al., 2013a). In the revision, we will provide a better description of the soil properties for TSP referring to detailed data from previous studies (Sørbotten, 2011; Wang et al., 2007; Zhu et al., 2013b).**

Minor comments:

Line 28 to 30: Nutrient imbalances in forest ecosystems and their effects on greenhouse gas emissions are very complex and shift in space and time. The present study analyzed effects of a single dose of P fertilizer on trace gas fluxes in an approximately 50-years old secondary forest over a period of 18 moths. It is simply too daring to extrapolate results from this special forest site to acid forest soils in general. Please do not speculate so much.

**R 1.4: Our results have potential implications for other N-saturated subtropical forests, despite that further tests are needed. We will rephrase our conclusions and largely confine them to our site.**

Line 127: change to: "In each block, plots were randomly assigned to a reference (Ref) and a P treatment."

**R 1.5: Agreed**

Line 164-165: How often and when did you measure trace gas fluxes? Did you take water samples (NH4+ and NO3- concentrations) at the same time? A different sampling time may explain the lack of correlation between N2O emissions and NO3- concentrations.

**R 1.6: Gas flux measurements were conducted monthly in the warm season and bi-monthly in the cold seasons. Soil water samples were retrieved on the same day the gas fluxes were measured and analyzed for $NO_3^-$ and $NH_4^+$. A lack of correlation between $N_2O$ emissions and $NO_3^-$ concentrations has been observed in other studies at TSP, e.g. by Zhu et al. (2013b), who attributed this finding the non-limiting $NO_3^-$ concentrations in the well-drained hillslope soils.**

Line 251-254: Why do you use different units (µg N m-2 hr-1; kg N ha-1 yr-1) for the same variable?

**R 1.7: We will revise the units, but stick to µg $N_2O$-N $m^{-2}$ $h^{-1}$ for instantaneous fluxes (e.g. Figs. 2 and 5) and kg $N_2O$-N $ha^{-1}$ $yr^{-1}$ for cumulative fluxes, as common in the GHG literature.**

Line 282: change to atmospheric

**R 1.8: OK.**

Line 362: Your study does not demonstrate that chronically high N deposition has transformed TSP soil to a regional hotspot for N2O and CH4 emissions. You have not measured that.

**R 1.9: This has been discussed in R 1.3 and previously demonstrated by Zhu et al. (2013b).**

Figures: Please provide information about sample size and if you use SD or SE in all of your figures.

**R 1.10: OK.**

Reference: Jones et al. (2016) Biogeosciences, 13, 4151–4165   Koehler et al. (2009)   Global Change Biology, 15, 2049 -2066

**Reference 1**

Chen, X. and Mulder, J.: Indicators for nitrogen status and leaching in subtropical forest ecosystems, South China, Biogeochemistry, 82(2), 165–180, doi:10.1007/s10533-006-9061-3, 2007.

Huang, Y., Kang, R., Mulder, J., Zhang, T. and Duan, L.: Nitrogen saturation, soil acidification, and ecological effects in a subtropical pine forest on acid soil in southwest China, J. Geophys. Res. Biogeosciences, 120, 2457–2472, doi:10.1002/2015JG003048., 2015.

Liu, B., Frostegård, Å. and Bakken, L.: Impaired Reduction of N2O to N2 in Acid Soils Is Due to a Posttranscriptional Interference with the Expression of nosZ, MBio, 5(3), 1383–14, doi:10.1128/mBio.01383-14.Editor, 2014.

Sørbotten, L.: Hill slope unsaturated flowpaths and soil moisture variability in a forested catchment in southwest China, Master Thesis, Nor. Univsity Life Sci. [online] Available from: http://brage.bibsys.no/xmlui/handle/11250/189382 (Accessed 22 September 2014), 2011.

Wang, Y., Solberg, S., Yu, P., Myking, T., Vogt, R. D. and Du, S.: Assessments of tree crown condition of two Masson pine forests in the acid rain region in south China, For. Ecol. Manage., 242(2–3), 530–540, doi:10.1016/j.foreco.2007.01.065, 2007.

Zhu, J., Mulder, J., Solheimslid, S. O. and Dörsch, P.: Functional traits of denitrification in a subtropical forest catchment in China with high atmogenic N deposition, Soil Biol. Biochem., 57, 577–586, doi:10.1016/j.soilbio.2012.09.017, 2013a.

Zhu, J., Mulder, J., Wu, L. P., Meng, X. X., Wang, Y. H. and Dörsch, P.: Spatial and temporal variability of $N_2O$ emissions in a subtropical forest catchment in China, Biogeosciences, 10(3), 1309–1321, doi:10.5194/bg-10-1309-2013, 2013b.

---

## Author Comment (AC2) · 26 Jan 2017

Dear Editor,

Please find below our response to the referees' comments on the manuscript "Phosphorus addition mitigates N2O and CH4 emissions in N-saturated subtropical forest, SW Chna". We would like to thank anonymous referees for their valuable and detailed comments. The answers are presented in bold and red fonts.

We also thank the editor for handling our manuscript. We remain at your disposal for any further enquiries.

Sincerely, Longfei Yu

Please also note the supplement to this comment:
http://www.biogeosciences-discuss.net/bg-2016-470/bg-2016-470-AC2-supplement.pdf

**Supplement:**

Received and published in BG Discussion on 03 Nov. 2016.

**Referee #2**

General comments The authors reported en experiment of P addition (79 kg P ha-1 yr-1, applied as NaH2PO4 powder) to an N saturated, Masson pine-dominated forest at TieShanPing (TSP), Chongqing, SW China for a period of 18 months. In the experiment, they measured soil fluxes of N2O and CH4, soil chemistry and plant growth. They found that P addition significantly decreased soil N2O emissions and turned the soil from CH4 emissions into a net sink. The experiment is appropriate. Data interpretation was logical and supported their conclusion. The study is for a type of ecosystem (subtropical, high ambient N deposition, N-saturated forest soil) for which such information is lacking. Their findings are interesting, can help us understand the interaction effect of N and P on greenhouse gas emission and also have implication for the forest management (such as P fertilization). In addition, the manuscript is also well organized and well written. I have only some minor concerns which I would like to discuss with the authors or maybe helpful for improving the manuscript, please see details below.

**R 2.0: We thank referee #2 for the positive feedback. The comments are addressed point by point below.**

Specific comments

1) P7, Line117. "annual N deposition at TSP measured in throughfall varies between 40-65 kg ha-1". What is the N deposition in precipitation there? Because N deposition in throughfall may be affected by difference in species, structure etc. of the forest, N deposition in precipitation is better data for comparing different forests.

**R 2.1: The bulk deposition at TSP is in a range of 20 to 30 kg N ha$^{-1}$ yr$^{-1}$ (Chen and Mulder, 2007). We will add this to our site description.**

2) P9, L165. Please give the locations (in the center of the plot?) where you measured the gas emission in the plots.

**R 2.2: Chambers for flux measurements were deployed next to the lysimeters, which were randomly distributed near the center in each of the 20m \* 20 m plots, at least 3 meters away from the border.**

3) P9, L165-167. "to investigate the immediate effect of P addition on ...... (7, 10 and 12 May) after the P application." Did you show these results in the result or discussion section? If not, please delete it.

**R 2.3: Yes, the data were shown in Fig. S6 and the result section (line 228). Other studies have reported stimulation of N$_2$O emission by P addition, presumably due to microbial response in soil (Mori et al., 2013; Wang et al., 2014). Hence, we included the short-term data for comparison.**

4) P11, Line 195 Statistical analyses. In the Experimental Design, the author showed that three blocks were established and two plots in each block in the study forest. In each block, plots were assigned randomly to a

reference (Ref) and a P treatment. Did you try One-way Repeated-Measures ANOVA to exam the treatment effect for the emission of CH4 and N2O, due to measuring the gas repeatedly.

**R 2.4: We have used repeated measures ANOVA to compare the fluxes of $CH_4$ and $N_2O$ as well as DIN concentrations among all our plots (Ref-1, Ref-2, Ref-3, P-1, P-2 and P-3). These six plots were compared as independent groups instead of three Ref plots as a group and three plots as the other. If we use $N_2O$ fluxes as an example, our results showed that in blocks 2 and 3, ref-2 and ref-3 were significantly larger than P-2 and P-3, respectively. Indeed, we believe and appreciate the suggestion from referee #1, that Linear Mixed Effects Models are better for interpreting treatment effects in our study. In the revision, we will reanalyze our dataset and anchor our discussion within the outcome of a Mixed Effect Model. For comparing parameters for tree-growth and soil properties (single observations only), one-way ANOVA is used (as in the original manuscript).**

5) P19, L362-363. "Overall, our study demonstrates that chronically high N deposition has transformed TSP soils to a regional hotspot for N2O and CH4 emission." It is not clear for me. Could you explain it?

**R 2.5: For details, please refer to R 1.3 and R1.8.**

6) P25, L565-568, "Zhang et al., 2014. Responses of nitrous oxide emissions to nitrogen and phosphorus

......" has been published in Biogeosciences, please replace Biogeosciences Discuss.

**R 2.6: Thanks. We will change it in the new version of manuscript.**

7) P27, L585, Table 1. How did you get n=6? Did you mixed samples in each plot?

**R 2.7: Before P application, we sampled the soil (for background properties) three times within each plot. Since there was no treatment yet, we combined two plots in the same block for presentation of the data.**

8) P28, L591, Table 2. The yearly variation for some data is big. For example, PAL in the ref plots was 5.4 in Aug. 2013, but was 13.4 in Aug. 2015. Do you have any explanation for it?

**R 2.8: Every half year, we sampled soil in triplicates from each plot randomly. Significant spatial heterogeneity is common in surface soils in 20*20 $m^2$ plots, as the litterfall may directly affect surface soil composition.**

**Reference 2**

Chen, X. and Mulder, J.: Indicators for nitrogen status and leaching in subtropical forest ecosystems, South China, Biogeochemistry, 82(2), 165–180, doi:10.1007/s10533-006-9061-3, 2007.

Mori, T., Ohta, S., Ishizuka, S., Konda, R., Wicaksono, A., Heriyanto, J. and Hardjono, A.: Effects of phosphorus addition with and without ammonium, nitrate, or glucose on N2O and NO emissions from soil sampled under Acacia mangium plantation and incubated at 100 % of the water-filled pore space, Biol. Fertil. Soils, 49(1), 13–21, doi:10.1007/s00374-012-0690-5, 2013.

Wang, F., Li, J., Wang, X., Zhang, W., Zou, B., Neher, D. a and Li, Z.: Nitrogen and phosphorus addition impact soil N2O emission in a secondary tropical forest of South China., Sci. Rep., 4, 5615, doi:10.1038/srep05615, 2014.

---

## Author Comment (AC3) · 26 Jan 2017

Dear Editor,

Please find below our response to the referees' comments on the manuscript "Phosphorus addition mitigates N2O and CH4 emissions in N-saturated subtropical forest, SW Chna". We would like to thank anonymous referees for their valuable and detailed comments. The answers are presented in bold and red fonts.

We also thank the editor for handling our manuscript. We remain at your disposal for any further enquiries.

Sincerely, Longfei Yu

Please also note the supplement to this comment:
http://www.biogeosciences-discuss.net/bg-2016-470/bg-2016-470-AC3-supplement.pdf

**Supplement:**

Received and published in BG Discussion on 03 Nov. 2016.

**Referee #3**

This paper report the effects of P addition on leachate chemistry and gas exchange of N2O and CH4 in a high N deposition forest of the warm and humid part of China. The results are interesting and the paper easy to read. However, the statistical treatment is not ideal, since the repeated nature of the measurements seems not to have been considered in the model tests. I see this problem is well addressed by the other reviewers. This needs to be addressed although it will not change the outcome and major conclusions.

**R 3.0: We thank referee #3 for the constructive and detailed suggestions. Regarding the statistical methods, we will reanalyze the dataset with Linear Mixed Effects Model as proposed by referee #1, so as to improve and justify the interpretation of our results. Please refer to our response to referee #1 for more details on statistical analyses.**

Below is my mainly minor comments listed by line number

70: 'and' = 'but'

**R 3.1: OK.**

71: delete: 'even'

**R 3.2: OK.**

123: spell out what means PAl here at first appearance.

**R 3.3: $P_{AL}$ means "ammonium lactate-extractable P, a common method to determine plant available P". We will modify it in the revised manuscript.**

126: 5-m buffer is a bit narrow, but cannot be changed

**R 3.4: The designs of plot size and buffering strip were made with reference to Zheng et al. (2016) and Martinson et al. (2013). In line 126, the "5-m buffer strip separated the two plots in each block" actually means that 5-m buffer was included for each plot, thus resulting in 10-m distance between the borders of two neighboring plots. To avoid confusion, we will modify this sentence in the new version.**

127: 'ad' = 'at'   130: '.... in the TSP soil.'   141: 'Within each plot, three ceramic....'

**R 3.5: We appreciate and accept the linguistic corrections from referee #3. Changes will be made in the revision.**

145: 'winter' = 'dormant' or 'dormant and dry'?

**R 3.6: The "winter season" mainly refers to the "dormant and dry season". We will rephrase it accordingly.**

200: should be repeated ANOVA of some kind. It seems to me that the statistical analysis is not optimal.

**R 3.7: Please refer to R2.4 for details.**

206: check the subscript on P

**R 3.8: We will check through and unify all the subscripts to "$P_{AL}$".**

290: I am not in favor of discussing degrees of N-saturation; I would instead say 'DHSRB is less N-rich with lower inorganic N availability than TSP'

**R 3.9: We agree with it in general, but we will just use "less nitrate-leaching" to describe DHSRB in comparison to TSP.**

303: 'frequently' = 'shortly'

**R 3.10: We assume that it refers to "frequently" in line 301 instead. We agree that "shortly" is more accurate.**

306: delete 'TSP' here, implied in nearby

**R 3.11: Thanks for the suggestion. We will delete "TSP".**

312-318: inhibition by NH4 cannot explain emission only lower uptake rates; so delete or reformulate

**R 3.12: We are aware that the inhibition by ammonium affects gross methane uptake and not directly net emissions. However, the observed mean $CH_4$ exchange rates (emission or uptake) at our TSP site was significantly smaller than reported in other subtropical forests from South China (Fang et al., 2009; Zhang et al., 2008) (Figs. 5 and 6). Therefore, it is reasonable to suggest that the inhibition of methane uptake by ammonium may have contributed to reverse net methane uptake to emission during "hotspots or hot moments" (Megonigal and Guenther, 2008) of methane production.**

330: like line 290; reformulate

**R 3.13: Changes will be made as presented in R3.9.**

328-334: I would suggest that both reason (and others as well) may have contributed

**R 3.14: As presented in the introduction section (line 91-93), "whether P addition affects the methanotrophic community in soils directly or alleviates the $NH_4^+$-inhibition effect on $CH_4$ oxidation through enhanced N uptake" remains under debate (Veraart et al., 2015). In our case, we only have evidence for reduction in nitrate availability from soil water, supporting the "indirect" mechanism.**

351-353: I do not understand this; what is the 'tree biomass estimates' doing here?

**R 3.15: Other studies have documented that P limitation may restrict tree growth in Masson pine forests (Wang et al., 2007). Our hypothesis was that P addition may enhance tree growth and thus N uptake. As discussed in our manuscript, the tree biomass estimates show that no such effect occurred within two years after P addition**

353-356: Why not, this should be simple and not much effort?

**R 3.16: From the previous long-term study conducted at TSP forest (Huang et al., 2015), we have learned that the abundance of ground vegetation species is highly variable from year to year. This makes the evaluation of ground vegetation biomass really uncertain in a two-year scale (our study). In the long-term experiment, we have planned to include the measurements of ground vegetation.**

436-38: something wrong in this ref    587: add 'lactate'    589, 594+595 add these lines to the table legend

**R 3.17: We thank referee #3's efforts on our manuscript. We will revise the manuscript according to the reviewer's suggestions.**

**Reference 3**

Fang, Y., Gundersen, P., Zhang, W., Zhou, G., Christiansen, J. R., Mo, J., Dong, S. and Zhang, T.: Soil–atmosphere exchange of N2O, CO2 and CH4 along a slope of an evergreen broad-leaved forest in southern China, Plant Soil, 319(1–2), 37–48, doi:10.1007/s11104-008-9847-2, 2009.

Huang, Y., Kang, R., Mulder, J., Zhang, T. and Duan, L.: Nitrogen saturation, soil acidification, and ecological effects in a subtropical pine forest on acid soil in southwest China, J. Geophys. Res. Biogeosciences, 120, 2457–2472, doi:10.1002/2015JG003048., 2015.

Martinson, G. O., Corre, M. D. and Veldkamp, E.: Responses of nitrous oxide fluxes and soil nitrogen cycling to nutrient additions in montane forests along an elevation gradient in southern Ecuador, Biogeochemistry, 112(1–3), 625–636, doi:10.1007/s10533-012-9753-9, 2013.

Megonigal, J. P. and Guenther, A. B.: Methane emissions from upland forest soils and vegetation., Tree Physiol., 28, 491–498, doi:10.1093/treephys/28.4.491, 2008.

Veraart, A. J., Steenbergh, A. K., Ho, A., Kim, S. Y. and Bodelier, P. L. E.: Beyond nitrogen: The importance of phosphorus for CH4 oxidation in soils and sediments, Geoderma, 259–260, 337–346, doi:10.1016/j.geoderma.2015.03.025, 2015.

Wang, Y., Solberg, S., Yu, P., Myking, T., Vogt, R. D. and Du, S.: Assessments of tree crown condition of two Masson pine forests in the acid rain region in south China, For. Ecol. Manage., 242(2–3), 530–540, doi:10.1016/j.foreco.2007.01.065, 2007.

Zhang, W., Mo, J., Zhou, G., Gundersen, P., Fang, Y., Lu, X., Zhang, T. and Dong, S.: Methane uptake responses to nitrogen deposition in three tropical forests in southern China, J. Geophys. Res. Atmos., 113(11), 1–10, doi:10.1029/2007JD009195, 2008.

Zheng, M., Zhang, T., Liu, L., Zhu, W., Zhang, W. and Mo, J.: Effects of nitrogen and phosphorus additions on nitrous oxide emission in a nitrogen-rich and two nitrogen-limited tropical forests, Biogeosciences, 13, 3503–3517, doi:10.5194/bg-2015-552, 2016.

---

## Author Response (AR1)

Received and published in BG Discussion on 03 Nov. 2016.

**Referee #1**

General comments:

Yu and co-authors present an interesting dataset of nitrous oxide (N2O) and methane (CH4) flux, nitrate (NO3-) concentration and other ancillary measurements from a forest ecosystem P-addition experiment conducted in a N-saturated (N-deposition is about 40 to 65 kg ha-1 yr-1) secondary Masson pine-dominated forest at TieShanPing (TSP), Chongqing, SW China (developed after a clear cut about 50 years ago) over a period of 18 months. After a single dose of P (applied as solid NaH2PO4*2H2O at a rate of 79.5 kg P ha-1) was added, Yu and co-authors found out that both N2O and CH4 emissions and NO3- concentrations in soil water decreased following P-addition during the 18-months period. They speculate that P-addition may have stimulated mineral N uptake by P-limited plants or microbes leading to decreased NO3- concentrations and N2O emissions. Concomitantly, decreasing mineral N concentrations may have relieved N inhibition of microbial CH4 oxidation leading to decreased CH4 emissions. Spatial and temporal dynamics of nutrient imbalances and their effects on biogeochemistry in forest ecosystems are very complex and difficult to decipher but important to understand in order to predict impacts of global change processes on trace gas fluxes in forest ecosystems. The results from Yu and co-authors are very interesting and valuable. Nutrient-addition experiments in whole forest ecosystems are very difficult to conduct but their results are often much more realistic than countless laboratory experiments trying to mimic whole ecosystem conditions.

**R 1.0: We thank referee #1 for the appraisal and valuable comments intended to improve the presentation of the data. We have responded below to the comments.**

However, I have major concerns and recommend a reanalysis of the dataset:

1) I would suggest to reanalyze the whole dataset using linear mixed effects models (Koehler et al. 2009 and Jones et al. 2016) to account for repeated measurements (monthly measurements over 1.5 years) and for within-group variance of a stratification (block-design of the study) which has not been done so far.

**R 1.1: We agree with the referee that Linear Mixed Effects Models would be better to capture within-group variances, even though the overall result may not differ substantially. In the revision, we will document the details of our statistical tests for different parameters, and present our final outcomes with reanalyzed results.**

2) Please take into account that all of your replicates in your plot are pseudo-replicates because they depend on one single block (which is your true replicate). In total you have three independent samples for the P-addition plots and three independent reference samples. Consequently, the dataset and figures should be reanalyzed and presented appropriately. If the blocks are so heterogenous you may show patterns in different blocks or outliers but your whole discussion should focus on significant results of the reanalyzed dataset based on plot means and not subplots.

**R 1.2: We agree with the referee's suggestions and will change the manuscript accordingly.**

3) I doubt that chronic N deposition alone has transformed TSP soils to a regional hotspot for N2O and CH4 emissions. You have not measured that. Changes in soil bulk density/soil compaction following a clear-cut about 50 years ago may be even more important. Water-filled pore space or gravimetric water content in soils are major controllers of N2O and CH4 production. These variables are almost always measured in soil trace gas flux studies and highly depend on soil bulk density which has also not been measured or is not presented in the manuscript. Since there was a clear-cut at the TSP site about 50 years ago and the soil type at this site is a Haplic Acrisol, where clay translocation processes in mineral horizons form clay-enriched horizons. Such a soil is prone to soil compaction. Especially in these clay-enriched horizons soil compaction may lead to increased bulk densities, may promote oxygen limitation and therefore increase rates of microbial denitrification and methanogenesis that eventually may lead to net CH4 and N2O emissions from this forest site. It would be great if authors could provide data about soil bulk densities, soil properties in general and soil water content variables.

**R 1.3: N saturation, indicated by significant nitrate leaching from soils, has been previously confirmed at TSP, (Chen and Mulder, 2007; Huang et al., 2015; Zhu et al., 2013b). Accordingly, annual $N_2O$ fluxes previously observed at TSP were among the highest fluxes reported in literature (Zhu et al., 2013b). Chronic N deposition does not only increase soil N availability, but also contributes to soil acidification (Huang et al., 2016), which is believed to facilitate large $N_2O$ emission factors ($N_2O$ flux: total N input) (Liu et al., 2014). Therefore, based on the previous findings, our study mainly addresses the P-addition effect on GHG emissions at TSP.**

**We also agree with the referee that edaphic factors, specific to Haplic Acrisols, play a role in the $N_2O$ and $CH_4$ emissions in TSP. Though WFPS was not directly monitored throughout our observation, the range of WFPS and its effect on $N_2O$ emissions in TSP have been well presented and studied by Zhu et al (2013b). The clear cut of the forests in the 1950s may affect the edaphic conditions in clay-rich horizon in soil, but not likely for the organic top horizon, where microbial activities such as denitrification were found to be strongest (Zhu et al., 2013a). In the revision, we will provide a better description of the soil properties for TSP referring to detailed data from previous studies (Sørbotten, 2011; Wang et al., 2007; Zhu et al., 2013b).**

Minor comments:

Line 28 to 30: Nutrient imbalances in forest ecosystems and their effects on greenhouse gas emissions are very complex and shift in space and time. The present study analyzed effects of a single dose of P fertilizer on trace gas fluxes in an approximately 50-years old secondary forest over a period of 18 moths. It is simply too daring to extrapolate results from this special forest site to acid forest soils in general. Please do not speculate so much.

**R 1.4: Our results have potential implications for other N-saturated subtropical forests, despite that further tests are needed. We will rephrase our conclusions and largely confine them to our site.**

Line 127: change to: "In each block, plots were randomly assigned to a reference (Ref) and a P treatment."

**R 1.5: Agreed**

Line 164-165: How often and when did you measure trace gas fluxes? Did you take water samples (NH4+ and NO3- concentrations) at the same time? A different sampling time may explain the lack of correlation between N2O emissions and NO3- concentrations.

**R 1.6: Gas flux measurements were conducted monthly in the warm season and bi-monthly in the cold seasons. Soil water samples were retrieved on the same day the gas fluxes were measured and analyzed for $NO_3^-$ and $NH_4^+$. A lack of correlation between $N_2O$ emissions and $NO_3^-$ concentrations has been observed in other studies at TSP, e.g. by Zhu et al. (2013), who attributed this finding the non-limiting $NO_3^-$ concentrations in the well-drained hillslope soils.**

Line 251-254: Why do you use different units (μg N m-2 hr-1; kg N ha-1 yr-1) for the same variable?

**We will revise the units, but stick to $\mu g \ N_2O-N \ m^{-2} \ h^{-1}$ for instantaneous fluxes (e.g. Figs. 2 and 5) and $kg \ N_2O-N \ ha^{-1} \ yr^{-1}$ for cumulative fluxes, as common in the GHG literature.**

Line 282: change to atmospheric

**R 1.7: ok**

Line 362: Your study does not demonstrate that chronically high N deposition has transformed TSP soil to a regional hotspot for N2O and CH4 emissions. You have not measured that.

**R 1.8: This has been discussed in R 1.3 and previously demonstrated by Zhu et al. (2013b).**

Figures: Please provide information about sample size and if you use SD or SE in all of your figures.

**R 1.9: ok**

Reference: Jones et al. (2016) Biogeosciences, 13, 4151–4165  Koehler et al. (2009)  Global Change Biology, 15, 2049 -2066

**Referee #2**

General comments The authors reported en experiment of P addition (79 kg P ha-1 yr-1, applied as NaH2PO4 powder) to an N saturated, Masson pine-dominated forest at TieShanPing (TSP), Chongqing, SW China for a period of 18 months.  In the experiment, they measured soil fluxes of N2O and CH4, soil chemistry and plant growth. They found that P addition significantly decreased soil N2O emissions and turned the soil from CH4 emissions into a net sink. The experiment is appropriate. Data interpretation was logical and supported their conclusion. The study is for a type of ecosystem (subtropical, high ambient N deposition, N-saturated forest soil) for which such information is lacking. Their findings are interesting, can help us understand the interaction effect of N and P on greenhouse gas emission and also have implication for the forest management (such as P fertilization). In addition, the manuscript is also well organized and well written. I have only some minor concerns which I would like to discuss with the authors or maybe helpful for improving the manuscript, please see details below.

**R 2.0: We thank referee #2 for the positive feedback. The comments are addressed point by point below.**

Specific comments

1) P7, Line117. "annual N deposition at TSP measured in throughfall varies between 40-65 kg ha-1". What is the N deposition in precipitation there? Because N deposition in throughfall may be affected by difference in species, structure etc. of the forest, N deposition in precipitation is better data for comparing different forests.

**R 2.1: The bulk deposition at TSP is in a range of 20 to 30 kg N ha$^{-1}$ yr$^{-1}$ (Chen and Mulder, 2007). We will add this to our site description.**

2) P9, L165. Please give the locations (in the center of the plot?) where you measured the gas emission in the plots.

**R 2.2: Chambers for flux measurements were deployed next to the lysimeters, which were randomly distributed near the center in each of the 20m * 20 m plots, at least 3 meters away from the border.**

3) P9, L165-167. "to investigate the immediate effect of P addition on ...... (7, 10 and 12 May) after the P application." Did you show these results in the result or discussion section? If not, please delete it.

**R 2.3: Yes, the data were shown in Fig. S6 and result section (line 228). Other studies have reported stimulation of N$_2$O emission by P addition, presumably due to microbial response in soil (Mori et al., 2013; Wang et al., 2014). Hence, we included the short-term data for comparison.**

4) P11, Line 195 Statistical analyses. In the Experimental Design, the author showed that three blocks were established and two plots in each block in the study forest. In each block, plots were assigned randomly to a reference (Ref) and a P treatment. Did you try One-way Repeated-Measures ANOVA to exam the treatment effect for the emission of CH4 and N2O, due to measuring the gas repeatedly.

**R 2.4: We have used repeated measures ANOVA to compare the fluxes of CH$_4$ and N$_2$O as well as DIN concentrations among all our plots (Ref-1, Ref-2, Ref-3, P-1, P-2 and P-3). These six plots were compared as independent groups instead of three Ref plots as a group and three plots as the other. If we use N$_2$O fluxes as an example, our results showed that in blocks 2 and 3, ref-2 and ref-3 were significantly larger than P-2 and P-3, respectively. Indeed, we believe and appreciate the suggestion from referee #1, that Linear Mixed Effects Models are better for interpreting treatment effects in our study. In the revision, we will reanalyze our dataset and anchor our discussion within the outcome of a Mixed Effect Model. For comparing parameters for tree-growth and soil properties (single observations only), one-way ANOVA is used (as in the original manuscript).**

5) P19, L362-363. "Overall, our study demonstrates that chronically high N deposition has transformed TSP soils to a regional hotspot for N2O and CH4 emission." It is not clear for me. Could you explain it?

**R 2.5: For details, please refer to R 1.3 and R1.8.**

6) P25, L565-568, "Zhang et al., 2014. Responses of nitrous oxide emissions to nitrogen and phosphorus

......" has been published in Biogeosciences, please replace Biogeosciences Discuss.

**R 2.6: Thanks. We will change it in the new version of manuscript.**

7) P27, L585, Table 1. How did you get n=6? Did you mixed samples in each plot?

**R 2.7: Before P application, we sampled the soil (for background properties) three times within each plot. Since there was no treatment yet, we combined two plots in the same block for presentation of the data.**

8) P28, L591, Table 2. The yearly variation for some data is big. For example, PAL in the ref plots was 5.4 in Aug. 2013, but was 13.4 in Aug. 2015. Do you have any explanation for it?

**R 2.8: Every half year, we sampled soil in triplicates from each plot randomly. Significant spatial heterogeneity is common in surface soils in 20*20 m$^2$ plots, as the litterfall may directly affect surface soil composition.**

**Referee #3**

This paper report the effects of P addition on leachate chemistry and gas exchange of N2O and CH4 in a high N deposition forest of the warm and humid part of China. The results are interesting and the paper easy to read. However, the statistical treatment is not ideal, since the repeated nature of the measurements seems not to have been considered in the model tests. I see this problem is well addressed by the other reviewers. This needs to be addressed although it will not change the outcome and major conclusions.

**R 3.0: We thank referee #3 for the constructive and detailed suggestions. Regarding the statistical methods, we will reanalyze the dataset with Linear Mixed Effects Model as proposed by referee #1, so as to improve and justify the interpretation of our results. Please refer to our response to referee #1 for more details on statistical analyses.**

Below is my mainly minor comments listed by line number

70: 'and'= 'but'

**R 3.1: Ok.**

71: delete: 'even'

**R 3.2: Ok.**

123: spell out what means PAl here at first appearance.

**R 3.3: P$_{AL}$ means "ammonium lactate-extractable P, a common method to determine plant available P". We will modify it in the revised manuscript.**

126: 5-m buffer is a bit narrow, but cannot be changed

**R 3.4: The designs of plot size and buffering strip were made with reference to Zheng et al. (2016) and Martinson et al. (2013). In line 126, the "5-m buffer strip separated the two plots in each block" actually means that 5-m buffer was included for each plot, thus resulting in 10-m distance between the borders of two neighboring plots. To avoid confusion, we will modify this sentence in the new version.**

127: 'ad' = 'at'   130: '.... in the TSP soil.'   141: 'Within each plot, three ceramic....'

**R 3.5: We appreciate and accept the linguistic corrections from referee #3. Changes will be made in the revision.**

145: 'winter' = 'dormant' or 'dormant and dry'?

**R 3.6: The "winter season" mainly refers to the "dormant and dry season". We will rephrase it accordingly.**

200: should be repeated ANOVA of some kind. It seems to me that the statistical analysis is not optimal.

**R 3.7: Please refer to R2.4 for details.**

206: check the subscript on P

**R 3.8: We will check through and unify all the subscripts to "$P_{AL}$".**

290: I am not in favor of discussing degrees of N-saturation; I would instead say 'DHSRB is less N-rich with lower inorganic N availability than TSP'

**R 3.9: We agree with it in general, but we will just use "less nitrate-leaching" to describe DHSRB in comparison to TSP.**

303: 'frequently' = 'shortly'

**R 3.10: We assume that it refers to "frequently" in line 301 instead. We agree that "shortly" is more accurate.**

306: delete 'TSP' here, implied in nearby

**R 3.11: Thanks for the suggestion. We will delete "TSP".**

312-318: inhibition by NH4 cannot explain emission only lower uptake rates; so delete or reformulate

**R 3.12: We are aware that the inhibition by ammonium affects gross methane uptake and not directly net emissions. However, the observed mean $CH_4$ exchange rates (emission or uptake) at our TSP site was significantly smaller than reported in other subtropical forests from South China** (Fang et al., 2009; Zhang et al., 2008) **(Figs. 5 and 6). Therefore, it is reasonable to suggest that the inhibition of methane uptake by ammonium may have contributed to reverse net methane uptake to emission during "hotspots or hot moments" (Megonigal and Guenther, 2008) of methane production.**

330: like line 290; reformulate

**R 3.13: Changes will be made as presented in R3.9.**

328-334: I would suggest that both reason (and others as well) may have contributed

**R 3.14: As presented in the introduction section (line 91-93), "whether P addition affects the methanotrophic community in soils directly or alleviates the $NH_4^+$-inhibition effect on $CH_4$ oxidation through enhanced N uptake" remains under debate (Veraart et al., 2015). In our case, we only have evidence for reduction in nitrate availability from soil water, supporting the "indirect" mechanism.**

351-353: I do not understand this; what is the 'tree biomass estimates' doing here?

**R 3.15: Other studies have documented that P limitation may restrict tree growth in Masson pine forests (Wang et al., 2007). Our hypothesis was that P addition may enhance tree growth and thus N uptake. As discussed in our manuscript, the tree biomass estimates show that no such effect occurred within two years after P addition**

353-356: Why not, this should be simple and not much effort?

**R 3.16: From the previous long-term study conducted at TSP forest (Huang et al., 2015), we have learned that the abundance of ground vegetation species is highly variable from year to year. This makes the evaluation of ground vegetation biomass really uncertain in a two-year scale (our study). In the long-term experiment, we have planned to include the measurements of ground vegetation.**

436-38: something wrong in this ref    587: add 'lactate'    589, 594+595 add these lines to the table legend

**R 3.17: We thank referee #3's efforts on our manuscript. We will revise the maunscript according to the reviewer's suggestions.**

**Details of revision:**

**Line 4**: "Department of Environmental Sciences" changed to "Faculty of Environmental Sciences and Natural Resource Management".

**Line 17**: "soil chemistry" to "soil N and P"

**Line 28-30:** "Our study suggests that P fertilization of N-saturated, subtropical forest soils could mitigate GHG emissions in addition to alleviate nutrient imbalances and reduce losses of nitrogen through $NO_3^-$ leaching and $N_2O$ emission." changed to "Our study indicates that P fertilization of N-saturated, subtropical forest soils mitigates $N_2O$ and $CH_4$ emissions, in addition to alleviating nutrient imbalances and reducing losses of N through $NO_3^-$ leaching."

**Line 68-69**: Add "on $N_2O$ emissions" after "(3 to 5 years)"

**Line 71:** Delete "even".

**Line 83:** Delete "other"

**Line 101-102**: Delete "ambient" and delete "ii) to test whether P affects N cycling in a highly N-saturated forest and"

**Line 109-110:** "Having a monsoonal climate, TSP has a mean annual precipitation" to "TSP has a monsoonal climate, with mean annual precipitation"

**Line 112**: "The soil is a loamy yellow mountain soil" changed to "Soils are predominantly well-drained, loamy yellow mountain soil"

**Line 115-117:** Add "The soil bulk density of the O/A horizon (~5 cm) is about 0.75 g $cm^{-3}$. Soil water-filled pore space (10 cm) at TSP hilltop generally ranges from 50 to 70% (annual mean ~ 60%; Zhu et al., 2013b)."

**Line 119-120:** Add "while the annual bulk N deposition is from 20 to 30 kg $ha^{-1}$ (Chen and Mulder, 2007)".

**Line 122:** Add "decreased" before "abundance"

**Line 125:** Change "$P_{Al}$" to "ammonium lactate-extractable P".

**Line 128:** Change "near a hilltop" to "well-drained soils of"

**Line 129-130:** Change "A 5-m buffer strip separated the two plots in each block. In each block, plots were assigned ad random to a Reference and a P treatment" to "Adjacent plots were separated by at least 10-m buffer zone. In each block, plots were randomly assigned to a Reference and a P treatment".

**Line 138:** "Together with the addition of phosphate, the P-treated plots also received 59.0 kg ha$^{-1}$ of sodium (Na)."changed to "The addition of $NaH_2PO_4·2H_2O$ at the P-treated plots also resulted in an input of 59.0 kg ha$^{-1}$ of sodium (Na)."

**Line 144:** Change "triplicates of" to "three".

**Line 145:** Add "near plot centre" after "at 5-cm and 20-cm soils".

**Line 148:** Change "in the winter season" to "in the dry and dormant season".

**Line 167:** Delete "in micro-plots"

**Line 198-213:** Linear mixed effect models have been applied to analyze time-series data, to account for both repeated measurements and within-group variance of a stratification variable (block design). The fixed effect (treatment by P or not) is then tested by analysis of variance. The whole dataset has been reanalyzed with more appropriate statistical methods using R. The most important details are added to the text.

**Line 216:** "$P_{AL}$" to "$P_{Al}$".

**Results Section:** The descriptions of differences in mean $NO_3^-$ concentrations, $N_2O$ and $CH_4$ fluxes among three blocks are now removed (Line 223-224; Line 236-237; Line 246-248).

**Line 232:** Add "values of" before "up"

**Line 259:** Add "of $N_2O$ emission rates" after "range"

**Line 262:** Delete "up to"

**Line 266:** Change "High" to "Large"

**Line 277:** Change "The" to ", which attributed the", delete "was attributed to"

**Line 278:** Change "due to" to "as a consequence of"

**Line 284:** Delete "(DHSBR)"

**Line 292:** Change "DHSBR" to "Dinghushan"; same applied to **Line 299 and 301**

**Line 293:** "which is not strongly different from that" to "which is similar to the N deposition at our site"

**Line 292-295:** "By contrast, the DHSBR site in South China receives 36 kg of atmospheric N ha$^{-1}$ yr$^{-1}$, which is only slightly smaller than the N deposition at our site (Huang et al., 2015), and showed larger $N_2O$ emission rate than the Ecuadorian site ($\sim$ 0.88 kg N ha$^{-1}$ yr$^{-1}$ in the reference plot; Zheng et al., 2016)." changed to "By contrast, the Dinghushan site in South China receives 28 kg N ha$^{-1}$ yr$^{-1}$ through wet inorganic N deposition (Zheng et al., 2016), which is similar to the

N deposition at our site (Chen and Mulder, 2007b; Huang et al., 2015). They also observed larger $N_2O$ emission rates ($\sim 0.88$ kg N ha$^{-1}$ yr$^{-1}$ in the Reference plots) than in the Ecuadorian site."

**Line 301:** Change "less N-saturated" to "less N-rich".

**Line 312:** Change "frequently" to "shortly".

**Line 317:** "TSP" deleted.

**Line 320:** Change "for South Chinese forest soils" to "for well-drained, forest soils in South Chinese forest"

**Line 321-325:** Change "As $CH_4$ fluxes at our sites were not significantly correlated with climatic factors (Fig. S4c and d), $CH_4$ emissions cannot be explained by (transient) wet conditions." to "Since aerated upland soils typically provide favourable conditions for microbial $CH_4$ uptake (Le Mer and Roger, 2010), the net emission observed in our sites is unlikely due to enhanced production, but rather by supressed consumption."

**Line 329:** Change "our" to "the TSP"

**Line 341:** Add "soil water loss due to"

**Line 341-342:** Change "the high degree of N saturation" to "the strong N enrichment".

**Line 343:** Change "the" to "direct"

**Line 344:** Change "the methane monooxygenase enzyme (Veldkamp et al., 2013), rather than a direct P-stimulation" to "methane monooxygenase (Veldkamp et al., 2013), rather than to P-stimulation"

**Line 352:** "the toxicity threshold" to "toxicity"

**Line 352-354:** Change "The frequent precipitation in the humid forest of this study (Yu et al., 2016), both prior and following the addition of $NaH_2PO_4.2H_2O$ (Fig. 2), efficiently diluted and leached Na$^+$, thus minimizing toxic effects." to "Frequent precipitation at TSP (Yu et al., 2016), both prior and following the addition of $NaH_2PO_4.2H_2O$ (Fig. 2), apparently diluted and leached Na$^+$, thus preventing toxic effects."

**Line 357-358**: Add "which represent a major decline in mobile anions in the P-treated soils"

**Line 363-364:** Add "increase in tree growth, due to"

**Line 375-376:** "our study demonstrates that chronically high N deposition has transformed TSP soils to a regional hotspot for $N_2O$ and $CH_4$ emission" changed to "our study suggests that N-saturated TSP soils act as a regional hotspot for $N_2O$ and $CH_4$ emissions".

**Line 605:** Add "lactate".

**Figs. 1, 4, 6, S3 and S6:** More appropriate statistics have been used to compare the data for the Reference and P treatments. Data used in the box whisker plots are now derived from the averages of subplot triplicates.

**Added references**

[revised manuscript text omitted]

$P_{H2O}$ = Water-extractable P, $P_{Al}$ = Ammonium lactate-extractable P,

$Al_{ox}$ = Oxalate extractable Al, $Fe_{ox}$ = Oxalate extractable Fe, $P_{ox}$ = Oxalate extractable P.

[*] Data not available

**Table 2** Soil pH, C, N and P contents in the O/A horizon (0-3 cm) in the References (Ref) and P

treatments. Values are means and standard deviations in parenthesis (n = 9). P addition was conducted on 14/05/04, after the first two sampling dates.

| | | pH | Total C g kg$^{-1}$ | Total N g kg$^{-1}$ | C/N | P$_{Al}$ mg kg$^{-1}$ | Total P mg kg$^{-1}$ |
|---|---|---|---|---|---|---|---|
| 13/08/02 | Ref | 3.7 (0.1)$^{bc\dagger}$ | 8.3 (2.3)$^{ab}$ | 0.5 (0.1)$^{bcd}$ | 16.9 (1.1)$^{bcd}$ | 5.4 (1.4)$^{c}$ | 292 (46)$^{bc}$ |
| | P | 3.6 (0.1)$^{c}$ | 6.7 (2.0)$^{b}$ | 0.4 (0.1)$^{bd}$ | 17.1 (2.1)$^{bc}$ | 5.1 (1.3)$^{c}$ | 260 (70)$^{c}$ |
| 14/05/02 | Ref | 3.7 (0.1)$^{abc}$ | 12.2 (4.2)$^{a}$ | 0.9 (0.3)$^{a}$ | 13.7 (1.5)$^{e}$ | 19.0 (8.0)$^{c}$ | 336 (65)$^{bc}$ |
| | P | 3.8 (0.2)$^{abc}$ | 9.0 (3.5)$^{ab}$ | 0.7 (0.2)$^{abc}$ | 14.2 (2.8)$^{de}$ | 13.7 (5.2)$^{c}$ | 270 (72)$^{bc}$ |
| 14/05/10 | Ref | 3.8 (0.1)$^{abc}$ | 9.9 (2.1)$^{ab}$ | 0.7 (0.2)$^{ab}$ | 14.0 (0.7)$^{e}$ | 15.4 (7.0)$^{c}$ | 304 (49)$^{bc}$ |
| | P | 3.9 (0.3)$^{ab}$ | 8.0 (1.9)$^{ab}$ | 0.6 (0.1)$^{bcd}$ | 14.3 (1.3)$^{cde}$ | 174 (114)$^{a}$ | 572 (242)$^{a}$ |
| 14/12/02 | Ref | 3.8 (0.1)$^{abc}$ | 10.5 (3.6)$^{ab}$ | 0.7 (0.3)$^{ab}$ | 14.5 (1.3)$^{cde}$ | 14.2 (7.4)$^{c}$ | 328 (102)$^{bc}$ |
| | P | 3.9 (0.2)$^{abc}$ | 9.5 (2.1)$^{ab}$ | 0.7 (0.1)$^{abc}$ | 14.0 (0.8)$^{e}$ | 66 (24)$^{ab}$ | 442 (106)$^{ab}$ |
| 15/08/02 | Ref | 3.9 (0.2)$^{ab}$ | 8.3 (2.2)$^{ab}$ | 0.4 (0.1)$^{cd}$ | 20.5 (2.5)$^{a}$ | 13.4 (6.2)$^{c}$ | 291 (61)$^{bc}$ |
| | P | 4.0 (0.2)$^{a}$ | 6.5 (1.9)$^{b}$ | 0.3 (0.1)$^{d}$ | 19.7 (2.2)$^{ab}$ | 57 (36)$^{ab}$ | 383 (136)$^{bc}$ |

$^{\dagger}$ Different letters indicate significant differences ($p < 0.05$).

[Figure]

**Fig. 1** Box whisker plots of $NH_4^+$ (a) and $NO_3^-$ (b) concentration in soil water at 5- and 20-cm depths in the References and P treatments, throughout 1.5 years after the P addition; red dashed lines indicate mean values; different letters indicate significant differences ($p < 0.05$).

[Figure]

**Fig. 2** Daily mean air temperature and precipitation (a), and monthly mean N$_2$O fluxes (±SE) in
the References (Ref) and P treatments in each of the three blocks (b-d); the red vertical line gives
the date of P addition (4 May, 2014).

[Figure]

**Fig. 3** Cumulative N$_2$O emissions for three blocks in the References (Ref) and P treatments from
summer 2013 to autumn 2015; the red arrows refer to the date of P addition (4 May, 2014).

[Figure]

**Fig. 4** Box whisker plots for $N_2O$ fluxes in the Reference and P treatment throughout 1.5 years after the P addition; red dashed lines indicate mean values; different letters indicate significant difference ($p < 0.05$).

[Figure]

**Fig. 5** Monthly mean CH$_4$ fluxes (±SE) in the References (Ref) and P treatments for three blocks
(a-c); the horizontal broken line indicates zero flux the red vertical line refers to the date of P
addition.

[Figure]

**Fig. 6** Box whisker plots of $CH_4$ fluxes in the Reference and P treatment throughout 1.5 years after the P addition; red dash lines indicate mean values; the different letters indicate significant difference ($p < 0.05$).

---

## Referee Report (RR1)

The manuscript by Longfei Yu and others presents a replicated forest fertilization experiment in an acidified and N-saturated Masson pine-dominated forest at TieShanPing, SW China. The experiment tests the role of mineral P fertilization in regulating nitrous oxide and methane emissions (uptake). Researchers measured soil water $NO_3^-$ concentrations, $N_2O$ emissions, $CH_4$ emissions (uptake), forest productivity, litter fall, litter chemistry, and soil characteristics prior to and after a one-time fertilization with 79 kg P ha-1 (NaH2PO4).

The Author's found that P fertilization results in declines in soil NO3- and suppressed emissions of N2O, and CH4 (not immediately, but over the long term). With P addition TSP soils switched from a CH4 source to a sink. Elevated biomass production was not observed over the 18-month experimental period. However, understory biomass was not assessed. Based on these results, Authors hypothesized that P additions resulted in increase NO3- uptake by plants and microbes leaving less for denitrification. Also, P addition was thought to lessen the NH4+ inhibition of methane oxidation.

Overall comments:
This manuscript is well written and presents a topic that is of interest generally to the readers of *Biogeosciences*. I have a couple of concerns that should be addressed prior to publication. First, I feel that these results and their interpretation would be easier to follow if there were a set of explicitly stated hypotheses. There is one hypothesis stated in the Abstract (that concerns the results), but not in the main body of the manuscript.

Regarding the description of the experimental design and sampling, the description in the methods doesn't seem to reflect the data that is presented for N2O and CH4 (L167-179). From the methods, I gather that N2O and CH4 were measured a total of four times, but clearly more data points are presented. Please clarify in the text how frequently measurements were made over the entire experiment. Other clarifying points are made below in the line-by-line comments.

With the Results, at times the text is confusing because the treatment effects of P additions and the seasonal/temporal patterns are explained simultaneously. I would recommend some minor reorganizing of this information. Perhaps start with the overall seasonal patterns and then state the treatment effects or the opposite.

Line-by-line comments:
L16: Change GHG to green house gas
L18: If this is a single fertilizer event is it necessary to have the unit yr-1?
L20: Rephrase this sentence to read "We observed a significant decline in soil water NO3- concentrations (5 and 20 cm depths) and in soil N2O emissions following P addition."
L21: It is unclear if this number is the amount of reduction or if it represents the total emission. Please clarify

L23-24: The "As for N2O" is a confusing way to begin this sentence. Can you revise to something like "P addition significantly decreased CH4 emissions, turning TSP soils from a net source to a net sink." I'm sure the Authors will have a more eloquent way of conveying that message.

L26-27: It's my preference to put this caveat in the discussion or that it's rephrased. The current wording suggests that you measured understory and that there was an increase in understory biomass.

L48-49: 'frequently shifting aerobic conditions' is awkward please revise. Perhaps this is better put in terms of aerobic and anaerobic?

P4 L56: Consider changing 'mineral' to 'inorganic'

L100: The hypothesis is stated in the abstract but not the main text. Please include in text prior to the objectives.

L105: It's unclear why the study site name is in quotations.

L116: TSP hilltop is not intuitive. Please explain in text

L121-122: Can you state over what time period the decline in growth has occurred here?

L128: Rather than an *, please use ×

L141: Can you report the $Na^+$ concentrations of the Reference plots?

L157: Change (2 mm) to (2mm × 2mm)

L165: I think part of the instrument name is missing. Should this be 'inductively coupled plasma atomic emission spectroscopy'? For all makes/models of equipment here and throughout, please add the location information.

L171: Change to "...into 12 mL pre-evacuated glass vials... (Chromacol, UK)."

L172: I would recommend splitting this into another sentence: "Vials were over pressurized to avoid contamination during sample transport."

L173: Is 'Mixing ratio' what you mean or should this be 'Fluxes of ...' or 'Concentrations of...'

L190: Please specify if the same trees were measured at each time point, this is critical to the interpretation of these data.

L194: Rather than 'sum of precipitation' can this be termed 'daily total precipitation'? Please provide the time period over which precipitation and temperature were measured.

L198: I gather from the methods that gas samples were collected from August 2013 forward, but only during the month of May (2, 7, 10, and 12). This doesn't reflect all of the data points that are shown in Figures 2 and 5. I would insist that the Authors add clarity to the methods or only show data that were collected in this study.

L206: It is unclear if fluxes of 'litterfall' nutrients were scaled to the biomass production. Or was litter biomass a component of the overall biomass calculation?

L210-213: For tree growth, how were the 3 different time points treated? Please be specific.

L226: The phrase 'sum of charge of dissolved base cations is unclear', at any rate, it would be more appropriate to say that charge was significantly different between fertilized and unfertilized. I am curious if the 'charge' decreases in the P treatment because of the increase of Na+. Can you please address?

L232-234: Please report the block effect here.

L236-238: Rephrase to read: The P addition resulted in a 50% (average 3 kg N ha-1 yr-1) reduction of cumulative N2O emissions (Fig. 3). Please add +/- Stderror if it is available

L238: Change was to were.

L240: Was there a significant block effect that could be reported here?

L245: Should this unit be CH4-C here and throughout? Also can you add +/- Stderror here?

L250: What does 138 t ha-1 represent? Is it an average across both years and both treatments? I'm not sure how informative that is. Based on your supplemental data, it looks as if biomass was actually lower in the P addition treatment compared to the Reference treatment.

L252: The 500g needle weight does not need to be reported here.

L253: This sentence needs to be clarified to indicate the mechanism responsible for differences in needle chemical composition. "Linked" is vague.

L253: 'hardly' is a vague word, please replace.

L273: Change mineral to inorganic

L273: This paragraph is long and difficult to follow. I believe that the Authors could find a way to make it more streamlined and easier to follow.

L275-279: This sentence is complex and confusing. Please revise, as it seems to contradict your former statement.

L283-292: I think it would be better to put your study into context of others that used similar additions. Perhaps the reference to moderate P additions is a bit of a distraction. I would recommend revising to focus on more similar studies.

L306: Likewise, the point of this paragraph is not entirely clear. As well, it is unclear if the referenced studies are also covering the short-term (~10day) span of time that is referenced in this manuscript.

L324: Change production to 'CH4 production' just to be clear this isn't primary production

L353: Change apparently to 'may have'

L355: I liked the nice flow and organization of this paragraph!

L375-376: Is there a citation from your previous work that you can add here? As is, these data don't provide obvious evidence for this.

L377: Can this statement be qualified by stating 'to overall reduce'

L379: GHG is used here and in the abstract, but is not explicitly defined. Please do so.

L388: References are not alphabetized consistently.

Tables and Figures:

L603: Please change 'Background' to 'Ambient'

Table 1. Was 5.0 mg kg-1 the detection limit of the instrument for $P_{H2O}$? If so, please just indicate this in the footer of the table rather than dedicating an entire column to the < 5.0 information.

Table 2. Here and throughout, please be consistent that the P treatment is '+P'. Also it is unclear what the letters indicate in terms of significance. Should they not indicate significant differences among the Ref and +P treatment? Or is this across all

time points? If so, the analysis should more appropriately be a repeated measures analysis.

Figure 4 and Figure 6: The letters indicating significance are somewhat unnecessary here. The point could be made in either the figure legend or with an asterisk centered above the two boxes. In both figures, I would recommend adding the statistical test that you used.

Figure S6: Litter is spelled incorrectly in the axis title

---

## Author Response (AR2)

Dear Editor,

Thank you for considering the second revision of our manuscript "**Phosphorus addition mitigates N$_2$O and CH$_4$ emissions in N-saturated subtropical forest, SW China**" (bg-2016-470) by Yu et al., for publication in Biogeosciences.

We have revised our manuscript (R2), based on the reviewers' comments. In the point-to-point response, the changes refer to the line numbers in the manuscript version with marked-up changes (attached following the response).

We thank you again for handling our manuscript.

Best regards,
Longfei Yu on behalf of all coauthors
longfei.yu@nmbu.no

**Comments from Reviewer #2**

1) Scientific Significance:

The primary objective of this study is to determine how phosphorus availability influences the efflux of N2O and CH4 in a nitrogen-saturated pine-dominated subtropical forest. While we have a fair understanding how the availability of nitrogen influences these processes, we have a very poor understanding how phosphorus can directly, or indirectly influence greenhouse gas production. Prior research on this subject has been inconclusive and this study will help add to the body of literature to advance our understanding of soil nutrient availability and climate forcing.

2) Scientific Quality:

Much improve from the early versions – mixed model is the way to go. My primary concern is the low replication (just three 20 x 20 m plots for each treatment) and perhaps overall discussion needs to be tempered.

R: We thank the reviewer for the critical but constructive comment. Our study explores the effect of phosphorus (P) on N cycling in the N-saturated Tieshanping (TSP) forest (Huang et al., 2015). Indeed, our field observations indicate that P application significantly reduces $N_2O$ and $CH_4$ emissions from upland soils at TSP. We agree that the current field set-up has its limitation and that our findings cannot be readily scaled up. For this, further research is needed, including work at other sites. The experiment was done in triplicate, as more replicates were not feasible given the available man-power and budget. In terms of replication, our experiment, investigating GHG emissions, does not differ much from others, which also used 20 x 20 $m^2$ plots in triplicates (Muller et al., 2015; Martinson et al., 2013) or even 10 x 10 $m^2$ plots in triplicates (Zheng et al., 2016; Wang et al., 2014) for each treatment.

3) Presentation Quality:

The discussion needs minor revisions. A fair amount of the discussion is just a summary of the results with modest interpretation or explanation. While it is fine to report how these results are similar/dissimilar with other work, it needs to go beyond that and provide new insights and tieback with the present study. What do the result consistent with this study have in common, but uncommon with work that is inconsistent? More effort is needed in the discussion to explain the 'why' of the results. Certain interesting passages need to be developed (e.g. lines 313-315)

R: We thank the reviewer for the general suggestion. In the revised version, we now expand our explanation of the results. Major changes are made in three sections of the discussion:

1) P effect on soil $NO_3^-$ concentration (line 301-310): we attribute the attenuation of soil $NO_3^-$ concentration to P stimulation of N uptake by plants and soil microorganisms. Based on the reported increase of N mineralization and nitrification rates in a similar site from South China, we suggest that microbial uptake of N is less likely than the plant uptake. Although this is not supported by our measurement of tree biomass and foliar N during two years, we provide an alternative explanation, which is N uptake by understory vegetation.

2) Comparison of P effect on $N_2O$ emission with other studies (line 324-335): In comparison with TSP, the slow response of $N_2O$ emission to P addition at the Ecuadorian site is suggested to be a result of both low ambient N deposition and low P dose. The Chinese site (Dinghushan) receives similar inorganic N input via throughfall as TSP. But we suggest the Dinghushan site to be less N-rich than TSP, based on the lower KCl-extractable inorganic N content in soil and lower flux of N leaching at Dinghushan. This point is now clearly made in the discussion.

3) We improve our explanation of N inhibition effects on $CH_4$ uptake by elaborating on known $CH_4/NH_4^+$ substrate competition for methane monooxygenase (line 372-373) and $NO_3^-/NO_2^-$ toxicity for methanotrophs (line 377-382). In the discussion of P effect on $CH_4$ uptake, we focus mainly on the direct

P effects on methantrophs and indirect P effects by reducing inorganic N levels in soils (line 386-398). The indirect mechanism seems more likely, as support by our observation.

Line 363: I question that the study actual demonstrates that high deposition has shifted these soils to regional hotspots for N2O and CH4 production – this was not tested. Also, this is difficult to say without historical data or somehow experimental removing/reducing N deposition. Yes, adding P has reduced NO3 concentration, but this result is fairly modest, especially at 20 cm.

R: We agree with the referee that historic data on the relationship between atmogenic N deposition and N concentrations in soil water as well as emission rates of $N_2O$ and $CH_4$ would be needed, to ultimately prove the connection between elevated N deposition and $N_2O$ and $CH_4$ emissions in the sub-tropical forest region. As such data do not exist for TSP, our conclusions are solely based on comparison with similar research sites and on manipulation experiments at TSP. For example, Huang et al (2015) showed that a doubling of N deposition at TSP caused a doubling of N leaching (as nitrate), simultaneously in increasing $N_2O$ emission (Liu WJ, personal communication). Our manuscript adds further evidence to this, as we find that P addition causes a significant decline in $NO_3^-$ concentrations in soil water in the surface horizons (O/A and AB; Fig. 1), while emission rates of $N_2O$ and $CH_4$ decrease (Figs. 4 and 6). Also other studies have reported strong correlation between $NO_3^-$ concentration in soil water and $N_2O$ emission rates (Gundersen et al., 2012).

Earlier studies at the TSP site indicate chronically elevated N deposition levels, high inorganic N concentration in soil water and strong soil acidification (Zhu et al., 2013; Larssen et al., 2011; Huang et al., 2015). These factors contribute to enhanced $N_2O$ emission and reduced $CH_4$ uptake in soils (Liu et al., 2010; Le Mer and Roger, 2010). Among other studies in Southern Chinese forests (Tang et al., 2006, Fang et al., 2009; Zhang et al., 2008), our study reports the highest $N_2O$ fluxes and lowest $CH_4$ uptake rates (or even net emission).

In addition, our recent [15]N tracer study at TSP (Yu et al., 2017), has shown that the enriched [15]N signals in emitted $N_2O$ is identical to those in soil $NO_3^-$ extracted from O/A horizon rather than AB horizon. This denotes the importance of $NO_3^-$ availability in surface soils to $N_2O$ emission. Also for the $CH_4$ oxidation, the N inhibition effect should be most important in surface soil (Bodelier and Laanbroek, 2004), where methanotrophic activity is more active with more aerobic condition. Therefore, our observation of significant decreases in $NO_3^-$ concentration from O/A and AB horizons do support that P addition migrates the $N_2O$ and $CH_4$ emissions from soil.

**Comments from Reviewer #3**

The manuscript by Longfei Yu and others presents a replicated forest fertilization experiment in an acidified and N-saturated Masson pine-dominated forest at TieShanPing, SW China. The experiment tests the role of mineral P fertilization in regulating nitrous oxide and methane emissions (uptake). Researchers measured soil water NO3- concentrations, N2O emissions, CH4 emissions (uptake), forest productivity, litter fall, litter chemistry, and soil characteristics prior to and after a one-time fertilization with 79 kg P ha-1 (NaH2PO4).

The Author's found that P fertilization results in declines in soil NO3- and suppressed emissions of N2O, and CH4 (not immediately, but over the long term). With P addition TSP soils switched from a CH4 source to a sink. Elevated biomass production was not observed over the 18-month experimental period. However, understory biomass was not assessed. Based on these results, Authors hypothesized that P additions resulted in increase NO3- uptake by plants and microbes leaving less for denitrification. Also, P addition was thought to lessen the NH4+ inhibition of methane oxidation.

**Overall comments:**

This manuscript is well written and presents a topic that is of interest generally to the readers of Biogeosciences. I have a couple of concerns that should be addressed prior to publication. First, I feel that these results and their interpretation would be easier to follow if there were a set of explicitly stated hypotheses. There is one hypothesis stated in the Abstract (that concerns the results), but not in the main body of the manuscript.

R: Following the reviewer's recommendation, we have changed our objectives to more explicit hypotheses. We now present three key hypotheses: 1) P addition stimulates tree growth; 2) P addition decreases soil inorganic N availability and 3) P addition reduces $N_2O$ and $CH_4$ emissions (line 105-109).

Regarding the description of the experimental design and sampling, the description in the methods doesn't seem to reflect the data that is presented for N2O and CH4 (L167-179). From the methods, I gather that N2O and CH4 were measured a total of four times, but clearly more data points are presented. Please clarify in the text how frequently measurements were made over the entire experiment. Other clarifying points are made below in the line-by-line comments.

R: Thanks for the suggestion. To avoid confusion, we now describe the frequency of gas emission sampling together with the soil pore water sampling. In line 178-180, we add "The measurements were conducted bi-monthly in the dry and dormant season and monthly during the growing season, simultaneously with the sampling of soil pore water". In line 181, we put "also" before the description of "short-term" intensive sampling.

With the Results, at times the text is confusing because the treatment effects of P additions and the seasonal/temporal patterns are explained simultaneously. I would recommend some minor reorganizing of this information. Perhaps start with the overall seasonal patterns and then state the treatment effects or the opposite.

R: We appreciate reviewer #3's advice. We have now reorganized the results of $N_2O$ and $CH_4$ fluxes, by starting with a paragraph for seasonal/temporal patterns of fluxes from Reference plots (line 243-255). The P effect on $N_2O$ and $CH_4$ fluxes are then presented in another two paragraphs.

Line-by-line comments:

L16: Change GHG to green house gas

R: OK.

L18: If this is a single fertilizer event is it necessary to have the unit yr-1?

R: We have revised it as "Here, we report $N_2O$ and $CH_4$ emissions together with soil N and P data for a period of 18 months following a single P addition (79 kg P $ha^{-1}$, as $NaH_2PO_4$ powder)" (line 17-19).

L20: Rephrase this sentence to read "We observed a significant decline in soil water NO3- concentrations (5 and 20 cm depths) and in soil N2O emissions following P addition."

L21: It is unclear if this number is the amount of reduction or if it represents the total emission. Please clarify

R: Agree. This number for emission reduction is now deleted. See line 20-22.

L23-24: The "As for N2O" is a confusing way to begin this sentence. Can you revise to something like "P addition significantly decreased CH4 emissions, turning TSP soils from a net source to a net sink." I'm sure the Authors will have a more eloquent way of conveying that message.

R: It is now changed to "P addition significantly decreased $CH_4$ emissions and turned the soil from a net source into a net sink" (line 24-26)

L26-27: It's my preference to put this caveat in the discussion or that it's rephrased. The current wording suggests that you measured understory and that there was an increase in understory biomass.

R: This sentence is now rephrased, as "Within the 1.5 years after P addition, no significant increase of forest growth was observed and P stimulation of forest N uptake by understory vegetation remains to be confirmed" (line 27-30)

L48-49: 'frequently shifting aerobic conditions' is awkward please revise. Perhaps this is better put in terms of aerobic and anaerobic?

R: OK (line 50-53).

P4 L56: Consider changing 'mineral' to 'inorganic'

R: Done.

L100: The hypothesis is stated in the abstract but not the main text. Please include in text prior to the objectives.

R: Please refer to our earlier reply to the overall comments. See also line 105-109.

L105: It's unclear why the study site name is in quotations. L116: TSP hilltop is not intuitive. Please explain in text

R: The quotation is removed now. "TSP hilltop" refers to upland soils from the hillslopes at TSP, which has been previously documented by Zhu et al., 2013b. Here, we rephrase "at TSP hilltop" as "on the hillslopes" (line 123-124).

L121-122: Can you state over what time period the decline in growth has occurred here?

R: Wang et al. (2007) and Li et al. (2014) reported the decline in forest growth at TSP during 2001-2004 and 2004-2012, respectively. So we rephrase the sentence as "Strong soil acidification has been reported to cause severe decline in forest growth at TSP since 2001 (Li et al., 2014; Wang et al., 2007)" (line 129-131).

L128: Rather than an *, please use ×
R: OK. Change is made in line 136.

L141: Can you report the Na+ concentrations of the Reference plots? L157: Change (2 mm) to (2mm × 2mm)
R: We add "(0.52-1.31 mg $L^{-1}$)" for the reference plots (line 149). More specific values for different soil depths could be found in Table S2 of the support information.

L165: I think part of the instrument name is missing. Should this be 'inductively coupled plasma atomic emission spectroscopy'? For all makes/models of equipment here and throughout, please add the location information.
R: OK. See line 174-175.

L171: Change to "...into 12 mL pre-evacuated glass vials... (Chromacol, UK)." L172: I would recommend splitting this into another sentence: "Vials were over pressurized to avoid contamination during sample transport."
R: It is modified as "20 ml gas samples were injected into pre-evacuated glass vials (12 ml) crimp-sealed with butyl septa (Chromacol, UK), maintaining overpressure to avoid contamination during shipment", as shown in line 182-184.

L173: Is 'Mixing ratio' what you mean or should this be 'Fluxes of ...' or 'Concentrations of...'
R: Yes. Mixing ratio is the direct result that we obtain from gas analysis on gas chromatograph coupled with ECD. Mixing ratios of $N_2O$ and $CH_4$ are commonly expressed as ppb or ppm, similar to "concentration". For fluxes, we need further calculation based on the change of gas mixing ratio with time, as described later in the same paragraph.

L190: Please specify if the same trees were measured at each time point, this is critical to the interpretation of these data.
R: Yes, they had been marked since the first measurement. This is now explained (line 204).

L194: Rather than 'sum of precipitation' can this be termed 'daily total precipitation'? Please provide the time period over which precipitation and temperature were measured.
R: OK (line 207-208).

L198: I gather from the methods that gas samples were collected from August 2013 forward, but only during the month of May (2, 7, 10, and 12). This doesn't reflect all of the data points that are shown in Figures 2 and 5. I would insist that the Authors add clarity to the methods or only show data that were collected in this study.

R: Please refer to the earlier reply to the overall comments of the reviewer. Changes are made accordingly in line 178-180.

L206: It is unclear if fluxes of 'litterfall' nutrients were scaled to the biomass production. Or was litter biomass a component of the overall biomass calculation? L210-213: For tree growth, how were the 3 different time points treated? Please be specific.

R: The fluxes of litterfall were not used for tree biomass evaluation. Instead, we used allometric models (specifically for masson pine; Li et al. 2011 and Zeng et al. 2008) to estimate the tree biomass, based on measured diameters at breast height (DBH). We specify now that DBH is used for tree biomass estimation (line 205).

The data for tree growth refer to Table S3, which includes tree biomass, 500-needle weight and needle nutrient contents. The three samplings were tested for treatment effect separately with one-way ANOVA (line 223-226).

L226: The phrase 'sum of charge of dissolved base cations is unclear', at any rate, it would be more appropriate to say that charge was significantly different between fertilized and unfertilized. I am curious if the 'charge' decreases in the P treatment because of the increase of Na+. Can you please address?

R: We thank the reviewer for the suggestion. This phrase is changed to "the overall cationic charge" (line 240-241). The increase in total cationic charge due to $Na^+$ addition should be modest, while the charge of dissolved of $Ca^{2+}$ is far more important. P addition stimulated the uptake of N, thus resulting in a decline in soil $NO_3^-$ concentration. As $NO_3^-$ is the major anion in soil water, the decrease of $NO_3^-$ concentration leads to a decline in cation concentration (Figure S3). Thus, the observed decrease in cationic charge is a direct effect of the decline in mobile anions (line 409-411).

L232-234: Please report the block effect here. L240: Was there a significant block effect that could be reported here?

R: The block effects were clearly described in the sentences that follow, in line 247-249 for $N_2O$ and in line 250-254 for $CH_4$.

L236-238: Rephrase to read: The P addition resulted in a 50% (average 3 kg N ha-1 yr-1) reduction of cumulative N2O emissions (Fig. 3). Please add +/- Stderror if it is available

R: We have revised it to "The P addition resulted in a 50% (3 kg N ha$^{-1}$ yr$^{-1}$ on average) reduction of cumulative $N_2O$ emission (Fig. 3)" (line 257-259).

L238: Change was to were.

R: "effects" is changed to "effect" (line 259).

L245: Should this unit be CH4-C here and throughout? Also can you add +/- Stderror here?

R: The units for $N_2O$ and $CH_4$ fluxes have been checked throughout the manuscript and corrected as g $N_2O$-N m$^{-2}$ hr$^{-1}$ and g $CH_4$-C m$^{-2}$ hr$^{-1}$, respectively. For annual emission, they are described as kg $N_2O$-N ha$^{-1}$ yr$^{-1}$ and kg $CH_4$-C ha$^{-1}$ yr$^{-1}$. Here, the $CH_4$ flux data show a skewed distribution, thus having large standard deviations. Since the treatment effect on $CH_4$ fluxes had been tested for significance with mixed-effect models, it is not necessary to include standard error here.

L250: What does 138 t ha-1 represent? Is it an average across both years and both treatments? I'm not sure how informative that is. Based on your supplemental data, it looks as if biomass was actually lower in the P addition treatment compared to the Reference treatment.
R: Agree. This value denotes the mean for both years and in both treatments. Based on statistical results, P treatment exhibited no difference from the reference. In the text, we have now described the change as "insignificant" and deleted the value for tree biomass (line 271-272).

L252: The 500g needle weight does not need to be reported here.
R: Agree.

L253: This sentence needs to be clarified to indicate the mechanism responsible for differences in needle chemical composition. "Linked" is vague.
R: It is rephrased as "Between the two samplings in 2013 and 2014, we found differences in chemical composition of the pine needles, but the difference between the Reference and P treatment was not significant" (line 273-275).

L253: 'hardly' is a vague word, please replace. L273: Change mineral to inorganic
R: OK. "hardly" is changed to "not" now (line 276).

L275-279: This sentence is complex and confusing. Please revise, as it seems to contradict your former statement.
R: OK. We have simplified our discussion, with additional details for support. "These findings are consistent with a number of previous studies (Baral et al., 2014; Hall and Matson, 1999; Mori et al., 2014), which attributed the reduction of $N_2O$ emissions in P-treated soils to decreased $NO_3^-$ availability and thus less denitrification. The attenuation of soil $NO_3^-$ by P addition at TSP may reflect stimulated N uptake by plants and/or soil microorganisms." (299-303).

L273: This paragraph is long and difficult to follow. I believe that the Authors could find a way to make it more streamlined and easier to follow.
R: See the next comment.

L283-292: I think it would be better to put your study into context of others that used similar additions. Perhaps the reference to moderate P additions is a bit of a distraction. I would recommend revising to focus on more similar studies.
R: This paragraph has now been divided into two paragraphs. The first one has been rephrased and focuses only on the key points of our experiment (296-310). The second one compares the fertilization effects of N, P and N+P on $N_2O$ emissions among forests with different N status, involving a space-for-time comparison. While our main discussion still addresses the comparison to similar studies in Southern China, we have modified the comparison with the Ecuadorian study, and focus on different responses of $N_2O$ emissions to P addition in forests with different N status (315-345). See more details in our response to comments from Reviewer #2.

L306: Likewise, the point of this paragraph is not entirely clear. As well, it is unclear if the referenced studies are also covering the short-term (~10day) span of time that is referenced in this manuscript.

R: To make it more concise, we have rewritten this paragraph, starting with studies that found that P addition increases $N_2O$ emission. Then we discuss the possible mechanisms for P stimulation of $N_2O$ emissions, and emphasize that the P effect on denitrifier activity could be rather fast (Mori et al., 2013c). However, we did not see any P effect on $N_2O$ emission from the intensive observation shortly after P application at TSP. This may be attributed to large denitrification potentials in TSP soils, supported by a lab-incubation study (Zhu et al., 2013c). See the changes in line 346-359.

L324: Change production to 'CH4 production' just to be clear this isn't primary production
R: OK. See line 370.

L353: Change apparently to 'may have'
R: OK. See line 407.

L355: I liked the nice flow and organization of this paragraph!
R: Much appreciated.

L375-376: Is there a citation from your previous work that you can add here? As is, these data don't provide obvious evidence for this.
R: We have added Zhu et al. (2013b) as a reference (line 429). For more details, see our reply to reviewer#2's comments.

L377: Can this statement be qualified by stating 'to overall reduce'
R: Yes, it can. See the changes made in line 430.

L379: GHG is used here and in the abstract, but is not explicitly defined. Please do so. L388: References are not alphabetized consistently.
R: For GHG, please refer to our previous response to the comment on line 16. References have been checked and corrected as requested.

Tables and Figures:
L603: Please change 'Background' to 'Ambient'
R: OK. See line 675.

Table 1. Was 5.0 mg kg-1 the detection limit of the instrument for PH2O? If so, please just indicate this in the footer of the table rather than dedicating an entire column to the < 5.0 information.
R: OK. This is now changed (line 675-681).

Table 2. Here and throughout, please be consistent that the P treatment is '+P'. Also it is unclear what the letters indicate in terms of significance. Should they not indicate significant differences among the Ref and +P treatment? Or is this across all time points? If so, the analysis should more appropriately be a repeated measures analysis.
R: The different letters indicate significant difference between the Reference and P treatment (line 682-685). This is tested by one-way ANOVA for each sampling, which has been specified in statistics (line 223-226).

Figure 4 and Figure 6: The letters indicating significance are somewhat unnecessary here. The point could be made in either the figure legend or with an asterisk centered above the two boxes. In both figures, I would recommend adding the statistical test that you used.

R: We thank the reviewer's suggestion. We would prefer keeping both the letters indicating significance and adding statistical methods to the figure captions. See line 699 and line 707.

Figure S6: Litter is spelled incorrectly in the axis title

R: OK. Revised.

[revised manuscript text omitted]

 $P_{Al}$ = Ammonium lactate-extractable P,

$Al_{ox}$ = Oxalate extractable Al, $Fe_{ox}$ = Oxalate extractable Fe, $P_{ox}$ = Oxalate extractable P.

[φ] Water-extractable P was below a detection limit of 5 mg kg$^{-1}$, thus not presented in table,

[*] Data not available

**Table 2** Soil pH, C, N and P contents in the O/A horizon (0-3 cm) in the References (Ref) and P

treatments. Values are means and standard deviations in parenthesis (n = 9). P addition was conducted on 14/05/04, after the first two sampling dates.

| | | pH | Total C | Total N | C/N | $P_{Al}$ | Total P |
|---|---|---|---|---|---|---|---|
| | | | g kg$^{-1}$ | g kg$^{-1}$ | | mg kg$^{-1}$ | mg kg$^{-1}$ |
| 13/08/02 | Ref | 3.7 (0.1)$^{ab\dagger}$ | 8.3 (2.3)$^{ab}$ | 0.5 (0.1)$^{ab}$ | 16.9 (1.1)$^{b}$ | 5.4 (1.4)$^{a}$ | 292 (46)$^{ab}$ |
| | P | 3.6 (0.1)$^{b}$ | 6.7 (2.0)$^{b}$ | 0.4 (0.1)$^{b}$ | 17.1 (2.1)$^{ab}$ | 5.1 (1.3)$^{a}$ | 260 (70)$^{b}$ |
| 14/05/02 | Ref | 3.7 (0.1)$^{a}$ | 12.2 (4.2)$^{a}$ | 0.9 (0.3)$^{a}$ | 13.7 (1.5)$^{b}$ | 19.0 (8.0)$^{a}$ | 336 (65)$^{a}$ |
| | P | 3.8 (0.2)$^{a}$ | 9.0 (3.5)$^{ab}$ | 0.7 (0.2)$^{ab}$ | 14.2 (2.8)$^{ab}$ | 13.7 (5.2)$^{a}$ | 270 (72)$^{a}$ |
| 14/05/10 | Ref | 3.8 (0.1)$^{ab}$ | 9.9 (2.1)$^{a}$ | 0.7 (0.2)$^{ab}$ | 14.0 (0.7)$^{b}$ | 15.4 (7.0)$^{b}$ | 304 (49)$^{b}$ |
| | P | 3.9 (0.3)$^{a}$ | 8.0 (1.9)$^{a}$ | 0.6 (0.1)$^{b}$ | 14.3 (1.3)$^{ab}$ | 174 (114)$^{a}$ | 572 (242)$^{a}$ |
| 14/12/02 | Ref | 3.8 (0.1)$^{a}$ | 10.5 (3.6)$^{a}$ | 0.7 (0.3)$^{a}$ | 14.5 (1.3)$^{ab}$ | 14.2 (7.4)$^{b}$ | 328 (102)$^{b}$ |
| | P | 3.9 (0.2)$^{a}$ | 9.5 (2.1)$^{a}$ | 0.7 (0.1)$^{ab}$ | 14.0 (0.8)$^{b}$ | 66 (24)$^{a}$ | 442 (106)$^{ab}$ |
| 15/08/02 | Ref | 3.9 (0.2)$^{ab}$ | 8.3 (2.2)$^{ab}$ | 0.4 (0.1)$^{ab}$ | 20.5 (2.5)$^{a}$ | 13.4 (6.2)$^{b}$ | 291 (61)$^{a}$ |
| | P | 4.0 (0.2)$^{a}$ | 6.5 (1.9)$^{b}$ | 0.3 (0.1)$^{b}$ | 19.7 (2.2)$^{ab}$ | 57 (36)$^{a}$ | 383 (136)$^{a}$ |

[†] Different letters indicate significant differences between References and P treatments (p < 0.05).

[Figure]

**Fig. 1** Box whisker plots of $NH_4^+$ (a) and $NO_3^-$ (b) concentration in soil water at 5- and 20-cm depths in the References and P treatments, throughout 1.5 years after the P addition; red dashed lines indicate mean values; different letters indicate significant differences ($p < 0.05$).

[Figure]

**Fig. 2** Daily mean air temperature and precipitation (a), and monthly mean N$_2$O fluxes (±SE) in
the References (Ref) and P treatments in each of the three blocks (b-d); the red vertical line gives
the date of P addition (4 May, 2014).

[Figure]

**Fig. 3** Cumulative N$_2$O emissions for three blocks in the References (Ref) and P treatments from
summer 2013 to autumn 2015; the red arrows refer to the date of P addition (4 May, 2014).

[Figure]

**Fig. 4** Box whisker plots for N₂O fluxes in the Reference and P treatment throughout 1.5 years after the P addition; red dashed lines indicate mean values; linear mixed-effect models were used to test the P treatment effect; different letters indicate significant difference ($p < 0.05$).

[Figure]

**Fig. 5** Monthly mean $CH_4$ fluxes (±SE) in the References (Ref) and P treatments for three blocks
(a-c); the horizontal broken line indicates zero flux the red vertical line refers to the date of P
addition (4 May, 2014).

[Figure]

**Fig. 6** Box whisker plots of $CH_4$ fluxes in the Reference and P treatment throughout 1.5 years after the P addition; red dash lines indicate mean values; linear mixed-effect models were used to test the P treatment effect; the different letters indicate significant difference ($p < 0.05$).

---

## Author Response (AR3)

Dear Editor,

We are really happy to be informed that our manuscript is finally accepted for publication at Biogeosciences pending small corrections. Attached are the final files after correction, including the text file for final publication as well.

We thank you again for providing all the useful comments and helping us all the way through the revision process.
Best regards,
Longfei Yu on behalf of all coauthors
longfei.yu@nmbu.no